# ATTENTION-BASED ITERATIVE DECOMPOSITION FOR TENSOR PRODUCT REPRESENTATION

**Taewon Park**[1]**, Inchul Choi**[2,*]**, Minho Lee**[1,2,*]
[1]Kyungpook National University, South Korea
[2]ALI Co., Ltd., South Korea
{ptw7998, sharpic77, mholee}@gmail.com

## ABSTRACT

In recent research, Tensor Product Representation (TPR) is applied for the systematic generalization task of deep neural networks by learning the compositional structure of data. However, such prior works show limited performance in discovering and representing the symbolic structure from unseen test data because their decomposition to the structural representations was incomplete. In this work, we propose an Attention-based Iterative Decomposition (AID) module designed to enhance the decomposition operations for the structured representations encoded from the sequential input data with TPR. Our AID can be easily adapted to any TPR-based model and provides enhanced systematic decomposition through a competitive attention mechanism between input features and structured representations. In our experiments, AID shows effectiveness by significantly improving the performance of TPR-based prior works on the series of systematic generalization tasks. Moreover, in the quantitative and qualitative evaluations, AID produces more compositional and well-bound structural representations than other works.[1]

## 1 INTRODUCTION

Humans can understand the compositional properties of the surrounding world and, based on their understanding, systematically generalize over unfamiliar things. This systematic generalization ability is one of the main characteristics of human intelligence and also the central issue of deep neural network research. However, the systematic generalization performance of deep neural networks is still far from human-level generalization (Fodor & Pylyshyn, 1988; Lake & Baroni, 2018; Hupkes et al., 2020; O'Reilly et al., 2022; Smolensky et al., 2022). Therefore, to improve the generalization performance, researchers have integrated symbolic system methodologies, such as Tensor Product Representation (TPR) (Smolensky, 1990), into neural networks.

TPR is a general method that explicitly encodes the symbolic structure of data with distributed representations. It is constituted by the tensor product of *roles* vectors and *fillers* vectors, where each encodes structural information and content of data. For decoding, it obtains symbol information from the embedded representation by applying TPR decoding components, *unbinding operators*. In TPR-based neuro-symbolic approaches, deep neural networks learn to extract TPR components (*roles*, *fillers*, and *unbinding operators*) from the input data while training for TPR functions. Recently, TPR-based neuro-symbolic approaches have showed significant improvements in the generalization and capacity of neural networks (Schlag & Schmidhuber, 2018; Schlag et al., 2020; 2021; Shi et al., 2022). However, we find that these approaches still encounter challenges in achieving compositional generalization. This is likely attributable to their reliance on a simple MLP for decomposition, which may not be structured to learn the compositional nature of data. Therefore, when the model fails to sufficiently learn the systematic decomposition of TPR, it inevitably degrades whole TPR operations (Smolensky, 1990; Bae et al., 2022).

In another approach to systematic generalization, the attention mechanism is used with neural networks to capture the compositional properties and thus generate meaningful *objects* representations

---

*corresponding authors
[1]The code of AID is publicly available at https://github.com/taewonpark/AID

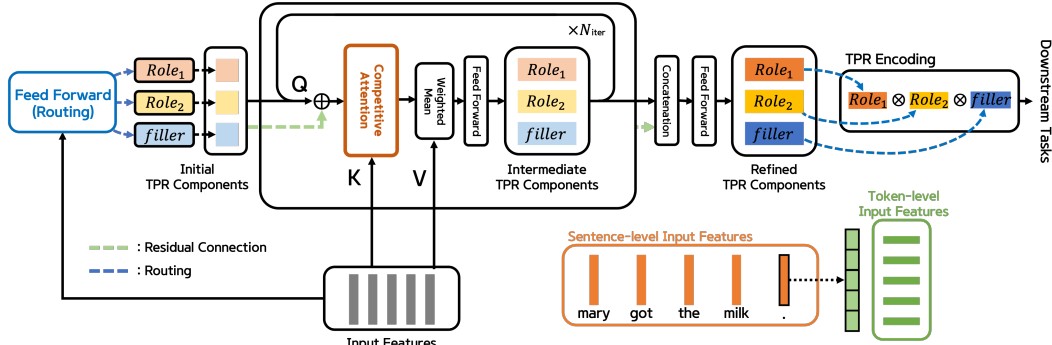

Figure 1: **Overview of AID assisted TPR decomposition.** We illustrate the overall operations of the AID-assisted part of the TPR-based model (with $N_{\text{inputs}} = 5$ and $N_{\text{com}} = 3$). In the natural language task, each input feature is a word vector for the sentence-level TPR models, while input features are $N_{\text{inputs}}$ sub-vectors partitioned from the word vector for the word-level TPR models. From those input features, the initial TPR components are obtained with a linear projection, and routed to structural representation slots of AID which iteratively decomposes TPR components with a competitive attention mechanism.

by selectively attending to relevant information (Goyal et al., 2019; Locatello et al., 2020). Such attention-based mechanisms have improved the sample efficiency and systematic generalization of neural networks in various fields, such as computer vision (Locatello et al., 2020; Singh et al., 2022) and reinforcement learning (Goyal et al., 2019; Yoon et al., 2023).

In this paper, to address the incomplete decomposition problem in TPR-based neural networks, we propose a novel Attention-based Iterative Decomposition (AID) module that can more effectively decompose sequential data into structured representations of TPR. The AID uses slot-based competitive attention (Goyal et al., 2019; Locatello et al., 2020; Singh et al., 2022) to bind sequential features to abstract structured representations (*roles* and *fillers*) of TPR, as shown in Fig. 1. While AID adopts an iterative competitive attention method similar to Slot Attention (Locatello et al., 2020), its distinctive contribution lies in its routing mechanism for the correct decomposition in the TPR framework. The TPR framework relies on pre-defined structural components, *roles* and *fillers*, to represent the underlying symbolic structure of data. To seamlessly integrate slot-based attention with the TPR framework, AID systematically routes individual structural components of TPR to a specific slot and refines the structured representation through competitive attention. Furthermore, AID conditionally initializes each slot component based on the context-dependent input features, instead of randomly initialized values in other slot-attention approaches. This learnable routing mechanism of AID provides a differentiable connection between input features and TPR components while learning to bind structured representations from input features in an end-to-end manner. AID can easily enhance the decomposition mechanism in any TPR-based model without introducing much computational overhead because of its parallel operations. Therefore, our method provides a simple and efficient way to enhance systematic generalization in any TPR-based models.

In experiments, we apply AID to several recent TPR-based or TPR equivalent approaches to show its effectiveness and flexibility. We adopt three types of TPR-related models for our study: (a) TPR-RNN (Schlag & Schmidhuber, 2018), (b) Fast Weight Memory (FWM) (Schlag et al., 2020), and (c) Linear Transformer (Katharopoulos et al., 2020). For comparison, we evaluate the AID-assisted version of all models with the baseline on the synthetic systematic generalization tasks, text/visual question-answering tasks, and even large-scale language modeling tasks. In all experimental results, AID shows effective generalization performance improvement for all TPR models.

**Our main contributions** are as follows: *(i)* we propose a novel AID module that can easily enhance the systematic generalization performance of any TPR-based models. *(ii)* we show that an iterative attention with routing mechanism effectively enhances structural representations of TPR (*roles*, *fillers*, and *unbinindg operators*) for unseen compositions in a systematic way. *(iii)* by improving the decomposition performance, we enable the application of TPR for large-scale real-world data.

## 2 RELATED WORK

**Binding problem** The binding problem, discussed by Greff et al. (2020), represents one of the neural network's abilities to flexibly and dynamically bind information from data to compositional structures. This problem comprises decomposing data into representations of meaningful entities and preserving their separation at the representational level to facilitate compositional reasoning (Bengio et al., 2013; Lake & Baroni, 2018; Goyal et al., 2019; Webb et al., 2020; Madan et al., 2021; Goyal & Bengio, 2022). TPR is one of the approaches for representational separation by explicitly binding *roles* and *fillers* representations using a tensor product. With TPR, this representational separation enhances the ability of neural networks to process the symbolic structure of data (Schlag & Schmidhuber, 2018; Schlag et al., 2019; 2020; Jiang et al., 2021; Shi et al., 2022). However, despite the binding capability of TPR, such TPR-based neuro-symbolic models do not explicitly learn the decomposition operations of TPR. Therefore, if they fail to decompose appropriate structural representations from unseen input data, it is likely to degrade all of the TPR functions of the network. To the best of our knowledge, our work represents the first attempt to address the decomposition problem in TPR-based neural networks.

**Compositional Generalization** The compositional generalization, also known as systematic generalization, has been explored in neural networks to enable them to generalize beyond what they learned (Fodor & Pylyshyn, 1988; Lake et al., 2017; Lake & Baroni, 2018; Liška et al., 2018; Hupkes et al., 2020; Webb et al., 2020). Recent research on compositional generalization has incorporated inductive biases, including attention mechanisms, into neural networks to capture self-contained, reusable, and disentangled representations (Goyal et al., 2019; Locatello et al., 2020; Mittal et al., 2021; Csordás et al., 2021; Madan et al., 2021; Singh et al., 2022). Among these approaches, Locatello et al. (2020) introduces an iterative attention-based mechanism called Slot Attention, designed to discover object representations within visual scenes. In contrast to Slot Attention, which assumes the permutation-invariant slots, our AID systematically routes each component to a specific structural component of TPR, such as *role* and *filler*. This routing strategy enables the generated representations to be utilized in TPR functions. In another line of research, Schlag et al. (2019); Jiang et al. (2021) explore the integration between attention and TPR, akin to our approach. However, they are designed for specific tasks such as math problems or abstractive summarization. In addition, Jiang et al. (2021) relies on a pre-defined *role* embedding dictionary. In contrast, the AID is a task-independent drop-in module that can be adapted to any TPR-based model. Also, it is designed to address a more fundamental problem, the decomposition of data into the appropriate *role* and *filler* simultaneously.

**Fast Weight Programmer** Our work is closely related to the field of Fast Weight Programmers (FWPs) (Schmidhuber, 1992; 1993). The concept of FWPs involves the construction of context-independent slow weights, which control context-dependent fast weights (Von Der Malsburg, 1994). During the training process, the slow weights are trained to program the context of input data effectively into the fast weights through back-propagation. This foundational concept has been extensively researched in prior work (Ba et al., 2016a; Schlag & Schmidhuber, 2017; 2018; Munkhdalai et al., 2019; Schlag et al., 2020; 2021; Irie et al., 2021). In FWPs, TPR is considered a high-order form of fast weight. In this context, our work is an extension of FWPs. In contrast to prior work, we focus on extracting fast weight representations from unseen data rather than designing updating rules for fast weight.

## 3 METHOD

In this section, we illustrate how the AID module decomposes input data into TPR components with iterative attention. Also, we detail the overall integration of the AID with pre-existing TPR-based models and show how it assists them.

### 3.1 TENSOR PRODUCT REPRESENTATION

We first briefly review TPR (Smolensky, 1990), which is a baseline method for our approach. TPR is a general method that explicitly encodes the symbolic structure of the objects in vector spaces. It is formed by an outer product between *role* and *filler* representations derived from the object.

Then, the connectionist representations for each object are combined via summation to represent multiple objects. During the decoding process, the *filler* is unbound from TPR by matrix multiplication with the *unbinding operator*. For the accurate encoding and decoding of symbolic structures in data, TPR requires three key conditions: (1) *roles* must be linearly independent of each other to prevent the overlap between *fillers*, (2) *unbinding operators* must exhibit a high correlation with the corresponding *roles* to access associated *fillers*, and (3) *fillers* encompass object information utilized for downstream tasks. These requirements inherently highlight the significance of decomposing *role/filler* representations from the input data to perform TPR functions (the conventional TPR relies on prior knowledge for *role/filler* decomposition). The outer product form of *roles* and *fillers* provides representational separation, which is necessary for addressing the binding problem (Greff et al., 2020). Based on this property, recent TPR-based models can enhance the generalization and interpretation of neural networks (Schlag & Schmidhuber, 2018; Schlag et al., 2019; 2020; Jiang et al., 2021; Shi et al., 2022). However, even in the framework of TPR, the representational separation is built upon the sufficiently learned *role/filler* decomposition in the neural network. Furthermore, our study finds that most TPR-based network models have difficulty when decomposing *role/filler* representations from unseen data. To address this problem, we introduce a slot attention-based iterative process (Locatello et al., 2020) for decomposition, which allows better generalization when dealing with unseen compositions of objects.

## 3.2 ATTENTION-BASED ITERATIVE DECOMPOSITION MODULE

AID is an iterative attention-based decomposition mechanism designed for TPR enhancement. At every time step, it takes input features and generates structured representations for TPR with competitive attention among intermediate TPR components, as illustrated in Algorithm 1. During the decomposition process, the AID can refine the structured representations with attention as often as $N_{\text{iter}}$ iterations. This attention-based competition mechanism enables AID to capture the compositional structure inherent in the data (Goyal et al., 2019; Locatello et al., 2020) and provides better generalization for unseen data.

For details, let us consider the AID module with a single iteration at time step *t*. At each time step, the AID takes $N_{\text{inputs}}$ input features and $N_{\text{com}}$ TPR components, denoted as `inputs` $\in \mathbb{R}^{N_{\text{inputs}} \times D_{\text{inputs}}}$ and `initial_components` $\in \mathbb{R}^{N_{\text{com}} \times D_{\text{com}}}$, and maps them to a common dimension of $D_{\text{com}}$ using learnable parameters **k**, **q**, and **v** for competitive attention. Subsequently, the AID computes an attention score denoted as `attn` $\in \mathbb{R}^{N_{\text{inputs}} \times N_{\text{com}}}$ through dot-product attention (Luong et al., 2015) between `key` and `query`. Based on `attn`, the AID applies a weighted mean to `value` to aggregate input information into `components` selectively, as follows.

$$\text{attn}_{i,j} := \frac{e^{A_{i,j}}}{\Sigma_l e^{A_{i,l}}} \quad \text{where} \quad A := \text{key} \cdot \text{query}^\top \in \mathbb{R}^{N_{\text{inputs}} \times N_{\text{com}}} \tag{1}$$

$$\text{updates} := W^\top \cdot \text{value} \in \mathbb{R}^{N_{\text{com}} \times D_{\text{com}}} \quad \text{where} \quad W_{i,j} := \frac{\text{attn}_{i,j}}{\Sigma_l \text{attn}_{l,j}} \tag{2}$$

During the attention process, each `component` competes with others to discover input features that better explain the symbolic structure for TPR. The AID adds the aggregated features to `components` for the update. According to Locatello et al. (2020), we apply layer normalization (LayerNorm) (Ba et al., 2016b) and multi-layer perception (MLP$_{\text{update}}$) of two layers with ReLU activation for the aggregated features. In the iterative process, the AID repeatedly refines those updated `components` using one layer MLP$_{\text{final}}$ and the final `components` are used to perform TPR functions for downstream tasks.

Notably, in its pure form, the iterative attention mechanism produces permutation-invariant `components`, posing a challenge in directly linking `components` to elements in TPR functions (e.g., identifying the appropriate `component` for *filler*). We introduce a trainable routing mechanism to our iterative attention method to integrate slot-based attention correctly with the TPR framework. In this method, each `initial_component` is systematically linked to specific symbols. As illustrated in Fig. 1, for instance, the first `component` is mapped to *role1* and the third to *filler*. Specifically, these `initial_components` are obtained by applying a feed-forward network to the concatenated input features and are optimized to facilitate a systematic decomposition during training. Also, these context-dependent `initial_components` significantly enhance the

training stability of the AID module. Furthermore, we apply additional training techniques to improve performance, including incorporating a residual connection (He et al., 2016), concatenation with DropOut (Srivastava et al., 2014), and non-linear activations for dot-product attention. The details of ablation studies for these techniques and a comprehensive evaluation of their impact on the overall model's performance are illustrated in Appendix D.

### 3.3 TPR NETWORKS WITH AID

In this section, we describe how the AID module is adapted to other TPR-based network models: (a) TPR-RNN (Schlag & Schmidhuber, 2018), (b) Fast Weight Memory (FWM) (Schlag et al., 2020), and (c) Linear Transformer (Katharopoulos et al., 2020). In the integration process, we apply the AID for both the encoding and decoding component generations with parameter sharing. Furthermore, we maintain the original TPR functions of the baseline models; instead, we replace their decomposition mechanisms with our AID module to improve the effectiveness of the decomposition.

#### 3.3.1 TPR-RNN

TPR-RNN is a TPR-based memory network designed for sentence-level processing. This model aims to solve basic question-answering tasks (Weston et al., 2015). TPR-RNN takes a set of word vectors $x_t = \{x_t^1, ..., x_t^{N_{\text{inputs}}}\}$ at each time step and generates a sentence vector $s_t = \Sigma_{i=1}^{N_{\text{inputs}}} x_t^i \odot p^i$, where $p = \{p^1, ..., p^{N_{\text{inputs}}}\}$ denotes learnable position vectors and $\odot$ denotes an element-wise product. Subsequently, TPR-RNN generates `components` based on the sentence vector $s_t$. In alignment with prior work, we derive the `initial_components` based on the sentence vector $s_t$. The AID performs decomposition for $x_t$ and `initial_components` to generate `components`.

#### 3.3.2 FWM

Fast Weight Memory (FWM) is a TPR-based memory network for word-level processing. This algorithm is specifically designed to enhance the capacity for long sequential data. FWM takes an input vector $x_t$ and generates a hidden state $h_t$ using internal LSTM (Hochreiter & Schmidhuber, 1997) at each time step. Subsequently, FWM generates `components` based on the hidden state $h_t$. To incorporate the AID, we form multiple input features using modular approaches akin to Henaff et al. (2016); Li et al. (2018). We transform $x_t$ into $\hat{x}_t$ using a feed-forward network and partition it into $N_{\text{inputs}}$ sub-vectors, $\{\hat{x}_t^1, ..., \hat{x}_t^{N_{\text{inputs}}}\}$. Follwing this transformation, the LSTM processes each $\hat{x}_t^i$ independently with shared parameters and generates multiple hidden states $\hat{h}_t = \{\hat{h}_t^1, ..., \hat{h}_t^{N_{\text{inputs}}}\}$. In alignment with prior work, we derive the `initial_components` based on the concatenated hidden state. The AID performs decomposition for $\hat{h}_t$ and `initial_components` to generate `components`.

#### 3.3.3 LINEAR TRANSFORMER

Recent research by Schlag et al. (2021) has demonstrated the equivalence between TPR and linear attention mechanisms. Building upon this crucial insight, we integrate the AID into the Linear Transformer architecture to enhance the linear attention mechanism from a TPR perspective. Our work uses the AID to extract key and value vectors for multi-head attention, which can be considered TPR encoding components. The Linear Transformer takes an input vector $x_t$ at each position and subsequently generates `components` based on $x_t$. In alignment with prior work, we also derive the `initial_components` based on the $x_t$ while simultaneously constructing multiple input features for competitive attention. To elaborate further, We transform $x_t$ into $\hat{x}_t$ using a feed-forward network and partition it into $N_{\text{inputs}}$ sub-vectors, $\{\hat{x}_t^1, ..., \hat{x}_t^{N_{\text{inputs}}}\}$. The AID performs decomposition for $\hat{h}_t$ and `initial_components` to generate `components`.

## 4 EXPERIMENT

In this section, we evaluate the effectiveness and flexibility of our proposed AID module for both systematic generalization tasks and a large-vocabulary language modeling task. These experiments

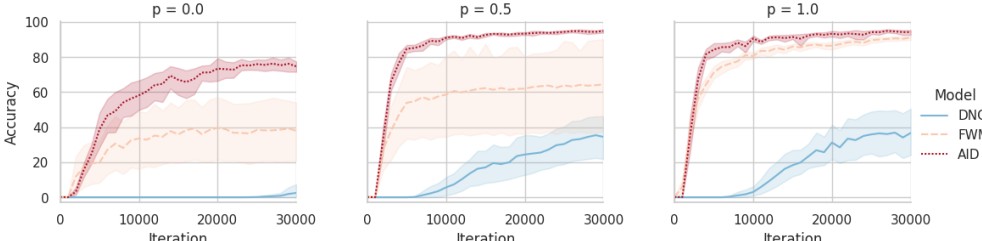

Figure 2: Test accuracy curve [%] on the SAR task for 10 seeds with varying values of $p$. As $p$ increases, the models can learn more combinatorial information of data during training.

show that the AID enhances the decomposition performance of TPR-based models and consequently provides overall systematic generalization performance improvement. For evaluation, we integrated the AID into established TPR-based approaches, as stated in Section 3.3. Also, we adopt the experimental configurations used in prior works for fair comparison. All experimental details and parameter settings are illustrated in Appendix B. We first perform a synthetically designed systematic generalization task (systematic associative recall) to show and verify the effectiveness of AID operations. Then, we perform the experiments on more real-world tasks such as systematic bAbI task, Sort-of-CLEVR task, and WikiText-103 task.

## 4.1 SYSTEMATIC ASSOCIATIVE RECALL TASK

To show the effectiveness of the AID module, we designed a new simplified Systematic Associative Recall (SAR) task that can measure the systematic generalization performance for unseen combinatorial sequence data. Because the conventional associative recall tasks do not adequately capture the systematic generalization of memory networks (Graves et al., 2014; Le et al., 2020; Park et al., 2021) or are complex to analyze (Bae et al., 2022), our SAR task mainly focused on assessing the model performance change according to varying compositional complexity of combinatorial input data during training. In this experiment, we adopted the FWM, a word-level TPR-based memory network, as our baseline model and compared it to other representative memory networks, including Differentiable Neural Computer (DNC) (Graves et al., 2016) and FWM.

**Task Description** The SAR task consists of two phases: a discovery phase involving memorizing input items and an inference phase recalling the memorized items. To introduce systematicity to the items, we design each item as a combination of $x \in X$ and $y \in Y$, where $X = X_1 \cup X_2 \cup X_3$ and $Y = Y_1 \cup Y_2$ represent independent word sets. The task provides combinatorial sequence data to a model during the discovery phase, and when the model is presented with an $x$, which is sampled from the sequence, it is required to recall the associated $y$ during the inference phase (we illustrate an example of the SAR task in Appendix B.1). We set different combination settings for each subset $X_i$ to evaluate the systematic generalization. During training, three types of combinations are provided to the models: (1) $X_1$ and $Y1$, (2) $X_2$ and $Y2$, and (3) $X_3$ and $Y$. In contrast, during evaluation, the models are supposed to memorize and recall unseen combinatorial data, specifically $X_1$ and $Y_2$. We also introduce a hyper-parameter $p$, which adjusts the proportion of types (2) and (3). Here, $p$ is defined as $\frac{|X_3|}{|X_2|+|X_3|}$, where $|X_i|$ denotes the cardinality of set $X_i$. A larger value of $p$ indicates that the models can learn more combinatorial information during training. By varying $p$, we can evaluate how the models' performance changes according to the diversity of the combinatorial items given during training time.

**Results** AID shows better generalization performance for all values of $p$, as shown in Fig. 2. As expected, the performance of the baseline models is degraded as the value of $p$ decreases, whereas the AID consistently shows high accuracy by successfully improving the baseline FWM.

### 4.1.1 ANALYSIS

We performed compositional analysis over the generated TPR component representations. For the quantitative aspect, disentanglement analysis is performed, and for the qualitative aspect, the orthogonality of representation is analyzed.

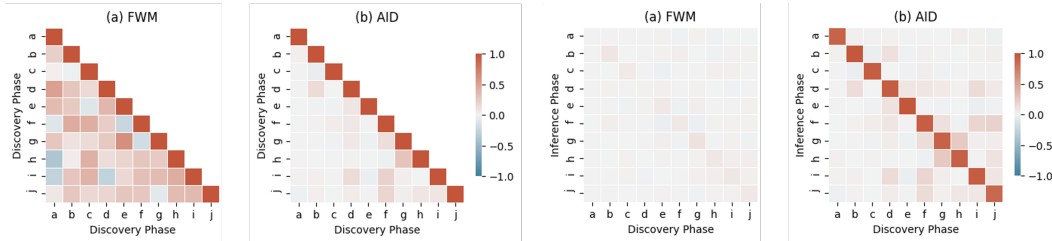

Figure 3: The heatmap for the cosine similarity between *roles* on the SAR task.

Figure 4: The heatmap for the cosine similarity between *roles* (x-axis) and *unbinding operators* (y-axis) on the SAR task.

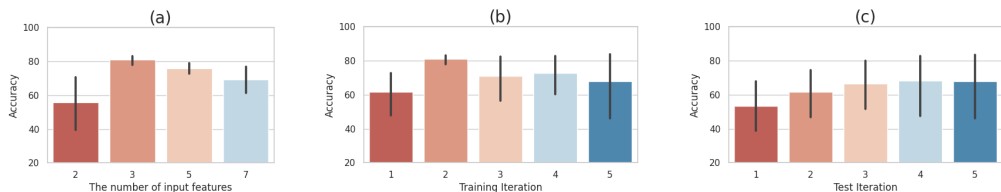

Figure 5: Test accuracy over different settings on the SAR task for 10 seeds. (a) Varying $N_{\text{inputs}}$ with $N_{\text{iters}} = 2$. (b) Varying training iterations with $N_{\text{inputs}} = 3$. (c) Varying test iterations for the models trained with $N_{\text{inputs}} = 3$ and $N_{\text{iters}} = 5$.

**Disentanglement Analysis**  We quantitatively evaluate the quality of disentanglement in the generated representations using the DCI framework (Eastwood & Williams, 2018). The DCI assesses the disentanglement (**D**), completeness (**C**), and informativeness (**I**) of the generated representations through the mapping from representations to generative factors (e.g., $x$ and $y$). Specifically, to evaluate multiple *role* and *filler* representations for each item, we employ a block-level DCI metric (Singh et al., 2022) by concatenating the TPR encoding components. Each metric indicates the degree to which the block representation disentangles the factors (**D**), how specialized each factor is to a specific block representation (**C**), and the accuracy of predicting ground-truth factor values based on the representations (**I**).

Table 1 shows the block-level DCI results for the baseline model and the AID across different $p$ values. Notably, the AID module enhances the quality of representational disentanglement and completeness compared to the baseline model across all $p$ values. These results demonstrate the AID module's efficacy in capturing underlying factors during TPR component generation, which may explain why the AID improves task performance.

Table 1: DCI results on the SAR task.

| Model | $p = 0.0$ | | | $p = 0.5$ | | | $p = 1.0$ | | |
|---|---|---|---|---|---|---|---|---|---|
| | **D** | **C** | **I** | **D** | **C** | **I** | **D** | **C** | **I** |
| FWM | 0.91 | 0.63 | 0.99 | 0.92 | 0.64 | 0.99 | 0.92 | 0.63 | 0.99 |
| + AID | 0.96 | 0.66 | 0.99 | **0.97** | **0.67** | 0.99 | **0.97** | **0.67** | 0.99 |

**Orthogonality Analysis**  Our qualitative analysis focuses on the generated representations in terms of orthogonality. As posited by Smolensky (1990), to ensure undistorted encoding and decoding processes, the *roles* should exhibit lower correlations among themselves, and the *unbinding operators* should exhibit high correlations with *roles* for the same $x$. To ascertain this, we evaluate the cosine similarity between TPR components for combinatorial data, with varying $x$ and fixed $y$, in the case of $p = 0.5$.

Fig. 3 shows the similarity between *roles* from the discovery phase, reflecting the quality of the encoding process. While the FWM yields disentangled representations, as indicated in Table 1, its *role* representations lack orthogonality for distinct $x$. In contrast, the AID generates highly orthogonal *role* representations. Fig. 4 shows the similarity between *roles* from the discovery phase and *unbinding operators* from the inference phase, reflecting the decoding process's quality. While the FWM generates less correlated *roles* and *unbinding operators*, even for the same $x$, and fails to solve the SAR problem, the AID generates highly correlated representations for the same $x$ and less correlated ones for distinct $x$. Furthermore, Figs. 3(b) and 4(b) show consistent patterns for *roles* and *unbind-*

Table 2: The mean word error rate [%] on the *sys-bAbI* task for 10 seeds.

| Model | | w/o sys diff | w/ sys diff | Gap |
|---|---|---|---|---|
| TXL | | $3.71 \pm 0.47$ | $8.72 \pm 3.27$ | 5.01 |
| DAM | | $0.48 \pm 0.20$ | $5.25 \pm 1.64$ | 4.77 |
| STM | | $0.49 \pm 0.16$ | $4.19 \pm 1.53$ | 3.7 |
| TPR-RNN | | $0.79 \pm 0.16$ | $8.74 \pm 3.74$ | 7.95 |
| | + AID | $0.69 \pm 0.08$ (0.10 ↓) | $5.61 \pm 1.78$ (3.13 ↓) | 4.92 (3.03 ↓) |
| FWM | | $0.79 \pm 0.14$ | $2.85 \pm 1.61$ | 2.06 |
| | + AID | $\textbf{0.45} \pm 0.16$ (0.34 ↓) | $\textbf{1.21} \pm 0.66$ (1.64 ↓) | $\textbf{0.76}$ (1.3 ↓) |

*ing operators*, indicating that the AID module correctly decodes the $filler$ information from the TPR-based memory. From all these results, it is clear that the AID module learns to decompose data into meaningful component representations that better conform to TPR conditions than the baseline model.

**The effect of $N_{\text{inputs}}$ and $N_{\text{iter}}$**  We analyze the effect of hyper-parameters, $N_{\text{inputs}}$ and $N_{\text{iters}}$, in the case of $p = 0.0$. In Fig. 5(a), we observe that the AID achieves its best performance when $N_{\text{inputs}}$ is set to 3. However, it shows slightly decreased performance at the other values of $N_{\text{inputs}}$. These results show an optimal number of input features for adequate competitive attention to decomposition. Fig. 5(b) shows that more iterations of decomposition enhance accuracy until a certain threshold, which aligns with the result from slot-attention(Locatello et al., 2020). When we increase $N_{\text{iters}}$ at test time, accuracy is increased until it reaches the training time iteration, as shown in Fig. 5(c).

## 4.2 Systematic bAbI task

Next, we evaluate the effect of the AID module on basic question-answering task, the bAbI task (Weston et al., 2015), which tests text understanding and reasoning abilities. To evaluate systematic generalization, we slightly modify the contents of the test data. This modified task, named *sys-bAbI*, evaluates the model's performance under two aspects: (i) in-distribution (*w/o sys diff*) and (ii) with the systematic difference (*w/ sys diff*). In this experiment, we use TPR-RNN and FWM as baseline models and compare our method to state-of-the-art memory networks, including Transformer-XL (TXL) (Dai et al., 2019), Distributed Associative Memory (DAM) (Park et al., 2021), Self-Attentive Associative Memory (STM) (Le et al., 2020), TPR-RNN, and FWM.

**Task Description**  The bAbI task consists of 20 distinct question-answering tasks, such as *time reasoning* and *yes/no questions*. Each task comprises stories, relevant questions, and corresponding answers. The bAbI task requires the models to remember the stories and recall information related to the questions to predict the correct answers. To evaluate systematic generalization, we modify the test data of the bAbI task by replacing words across sub-tasks. Specifically, we replace human names, such as *Mary*, *John*, and *Fred*, which appear in various tasks (alternative word types, such as location names, can also be replaced). Through this replacement, the test data of each task will contain words that are not visible during training. The models are expected to learn task-independent text understanding to solve the *sys-bAbI* task. We train the models on all tasks and evaluate them on subsets of the bAbI tasks that are reasonable for implementing the replacement.

**Results**  Table 2 shows the text understanding performance in *w/o sys diff* and *w/ sys diff* settings on the *sys-bAbI* task. Existing memory networks record a high-performance gap between both settings despite their solid achievement on the original test data, *w/o sys diff*. The AID improves the baseline models' systematic generalization and performs better than other memory networks in both settings. This result indicates that TPR-based approaches benefit systematic generalization but require a more elaborate design for decomposing structured representations.

## 4.3 Sort-of-CLEVR task

We evaluate the competence of the AID module in visual relational reasoning tasks. We use the Sort-of-CLEVR task (Santoro et al., 2017), which evaluates compositional generalization for visual scenes. For our experiments, we use Linear Transformer as the baseline model for comparison.

Table 3: Test accuracy [%] on the Sort-of-CLEVR task for 10 seeds.

| Model | Unary Accuracy | Binary Accuracy | Ternary Accuracy |
|---|---|---|---|
| Linear Transformer | 69.3 ± 14.8 | 75.5 ± 1.3 | 56.4 ± 4.3 |
| + AID | **98.9** ± 0.2 (29.6 ↑) | **78.6** ± 0.3 (3.1 ↑) | **63.7** ± 1.2 (7.3 ↑) |

Table 4: Perplexity on the WikiText-103 task. [†]The experimental results are obtained through open sources provided by (Schlag et al., 2021).

| Model | Valid | Test |
|---|---|---|
| Linear Transformer[†] | 36.473 | 37.533 |
| + AID | **36.159** (0.314 ↓) | **37.151** (0.382 ↓) |
| Delta Network[†] | 35.640 | 36.659 |
| + AID | **35.361** (0.279 ↓) | **36.253** (0.406 ↓) |

**Task Description** The sort-of-CLEVR task is a visual question-answering task that evaluates the visual understanding of objects. It includes scene images, relevant questions, and corresponding answers. Introduced by Mittal et al. (2021), this task consists of three question types: (i) the properties of single objects (*Unary*), such as color and shape, (ii) the relationship between two objects (*Binary*), and (iii) the relationship among three objects (*Ternary*). To successfully solve this problem, models must capture the properties of the objects or their relationships to predict the correct answers.

**Results** Table 3 shows the generalization performance on the sort-of-CLEVR task for different question types. Remarkably, the Linear Transformer struggles to solve these fundamental visual question-answering tasks. However, with the integration of the AID module, the baseline model's generalization performance improves significantly, demonstrating the effectiveness of the AID in enhancing the generation of the Linear Transformer's key and value vectors.

## 4.4 WikiText-103 task

In addition to systematic generalization tasks, we extend our evaluation to assess the effectiveness of the AID on a more practical large-scale task that may not explicitly demand systematic generalization capability but is an important problem. WikiText-103 task (Merity et al., 2016) comprises lengthy corpora from Wikipedia articles and evaluates the model's understanding of the probability distribution over a sequence of words through perplexity. We use Linear Transformer and Delta Network (Schlag et al., 2021) as our baseline models for this experiment and compare our method against them.

**Results** In the experiment, we use a 16-layered Linear Transformer and a 16-layered Delta Network as our baseline models, in accordance with Schlag et al. (2021). To mitigate the additional computational overhead by the AID, we strategically integrate our AID module into the baseline models at intervals of 4 out of the 16 layers. Table 4 shows the perplexity results of the WikiText-103 task. Notably, our AID improves the perplexity of baseline models in both baseline models, demonstrating its potential for enhancing performance even on large-scale tasks.

## 5 Conclusion

In this work, we propose a novel AID module that effectively decomposes sequential data into structured representations of TPR. AID iteratively applies slot-based competitive attention to bind sequential features to abstract structured representations of TPR and also learns to route specific symbolic meanings to each slot for the context-dependent initialization of *roles* and *fillers*. In experiments, we demonstrate the effectiveness of the AID module on synthetic systematic generalization tasks, text/visual question-answering tasks, and even large-scale language modeling tasks. Our analysis shows that the AID provides well-bound representations for TPR-based networks and successfully enhances overall systematic generalization performance.

ACKNOWLEDGEMENTS

This work was supported by the National Research Foundation of Korea(NRF) grant funded by the Korea government(MSIT)(No. 2022R1A5A7026673).

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

## A    PSEUDO-CODE

---

**Algorithm 1** Attention-based Iterative Decomposition module.

---

1: **Input:** inputs $\in \mathbb{R}^{N_{\text{inputs}} \times D_{\text{inputs}}}$, initial_components $\in \mathbb{R}^{N_{\text{com}} \times D_{\text{com}}}$
2: **Layer params: k, q, v**: linear projections for attention; $\text{MLP}_{\text{update}}$; LayerNorm ; $\text{MLP}_{\text{final}}$
3:    components = initial_components
4:    key, value = **k**(inputs), **v**(inputs)
5:    key = $\text{ELU}(\text{key}) + 1$
6:    **for** $n = 0...N_{\text{iter}}$ **do**
7:        query = **q**(components) + initial_components
8:        query = $\frac{1}{\sqrt{D_{\text{com}}}}$query
9:        query = $\text{ELU}(\text{query}) + 1$
10:        attn = $\text{Softmax}(\text{key} \cdot \text{query}^\top, \text{axis}=\text{`components'})$
11:        updates = $\text{WeightedMean}(\text{weight}=\text{attn}+\epsilon, \text{values}=\text{value})$
12:        components += $\frac{1}{D_{\text{com}}} \text{MLP}_{\text{update}}(\text{LayerNorm}(\text{updates}))$
13:    **end for**
       components = $\text{MLP}_{\text{final}}(\text{components}, \text{DropOut}(\text{initial\_components}))$
14:    **return** components

---

## B    EXPERIMENTAL SETTINGS

### B.1    SYSTEMATIC ASSOCIATIVE RECALL TASK

**Task Description**    The SAR task consists of two phases: a discovery phase involving memorizing input items and an inference phase recalling the memorized items. To introduce systematicity to the items, we design each item as a combination of $x \in X$ and $y \in Y$, where $X = X_1 \cup X_2 \cup X_3$ and $Y = Y_1 \cup Y_2$ represent independent word sets. The task provides combinatorial sequence data to a model during the discovery phase, and when the model is presented with an $x$, which is sampled from the sequence, it is required to recall the associated $y$ during the inference phase. We set different combination settings for each subset $X_i$ to evaluate the systematic generalization. During training, three types of combinations are provided to the models: (1) $X_1$ and $Y1$, (2) $X_2$ and $Y2$, and (3) $X_3$ and $Y$. In contrast, during evaluation, the models are supposed to memorize and recall unseen combinatorial data, specifically $X_1$ and $Y_2$.

Fig. 6 illustrates an example of the SAR task. At each training iteration, generative factors ($x$ and $y$) are sampled from word sets $X$ and $Y$ to construct the input sequence. The sampled $x$ and $y$ values are randomly paired, creating combinations of one $x$ and one $y$ each. These pairs are then embedded into a vector space and concatenated with flags, which are scalar values signaling the start of the discovery and inference phases. In the discovery phase, models sequentially receive these concatenated representations. During the inference phase, the model is presented only with $x$ values (considered as *role*) and is tasked with predicting the corresponding $y$ values (considered as *fillers*).

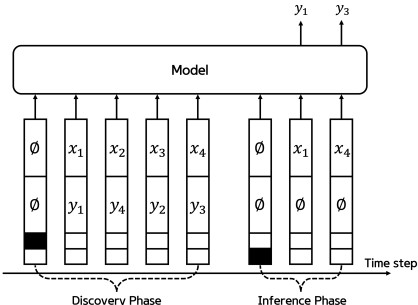

Figure 6: Example of the SAR task for sampled pairs $\{\{x_1, y_1\}, \{x_2, y_4\}, \{x_3, y_2\}, \{x_4, y_3\}\}$ from word sets $X$ and $Y$.

**Experiment Details**    We use a set of arbitrary 1,000 words to construct each word set, as outlined in Table 15. We map each word in a 50-dimensional space using the word embedding method. Therefore, each item is formed by concatenating the embedding vectors for $x$ and $y$. The experiments on the SAR task are repeated 10 times with distinct seeds. During training, we utilize the Adam optimizer with a batch size of 64 and a learning rate of $1e^{-3}$, $\beta_1$ of 0.9, and $\beta_2$ of 0.98 for training iterations of $30K$ (for training DNC, we use the RMSprop optimizer with a learning rate of $1e^{-4}$ and $\alpha$ of 0.99).

### B.2 SYSTEMATIC BABI TASK

**Task Description**    The bAbI task (Weston et al., 2015) consists of 20 distinct question-answering tasks, such as *time reasoning* and *yes/no questions*. Each task comprises stories, relevant questions, and corresponding answers. The bAbI task requires the models to remember the stories and recall information related to the questions to predict the correct answers. To evaluate systematic generalization, we modify the test data of the bAbI task by replacing words across sub-tasks. As shown in Table 5, we replace {*Daniel, John, Sandra*} to {*Bill, Fred, Julie*}, and vice versa, during the replacement process. Since this modification is applied to the subset of the bAbI tasks, we evaluate the models on the modified tasks while training them on all tasks.

**Experiment Details**    We use the experiment settings used in TPR-RNN (Schlag & Schmidhuber, 2018) and FWM (Schlag et al., 2020). The experiments on the *sys-bAbI* task are repeated 10 times with distinct seeds. We also use the word embedding method to embed words in vector spaces.

For TPR-RNN, we set the embedding size to 179. For training, we utilize the Adam optimizer with a batch size of 64 and a learning rate of $1e^{-3}$, $\beta_1$ of 0.9, and $\beta_2$ of 0.99 for 100 training epochs.

For FWM, we set the embedding size to 256. For training, we utilize the Adam optimizer with a batch size of 64 and a learning rate of $1e^{-3}$, $\beta_1$ of 0.9, and $\beta_2$ of 0.98 for training iterations of $60K$.

Additionally, we incorporate a reconstruction loss, following the approach employed in Park et al. (2021), to enhance representation learning on the *sys-bAbI* task. In our reconstruction loss, the models are required to predict words based on their word embedding vectors. This loss is scaled to 1/100th and added to the main task loss.

Table 5: Word replacement across sub-tasks on *sys-bAbI* task.

| Sub-task number | 1, 2, 3, 6, 7, 8, 9, 11, 12, 13 | | 10, 14 |
|---|---|---|---|
| Word | {*Daniel, John, Sandra, Mary*} | $\leftrightarrow$ | {*Bill, Fred, Julie, Mary*} |

### B.3 SORT-OF-CLEVR TASK

**Task Description**    The sort-of-CLEVR task is a visual question-answering task that evaluates the visual understanding of objects. It includes scene images, relevant questions, and corresponding answers. Each scene image contains 6 objects, each characterized by a distinct shape and color selected from 2 possible shapes (square or circular) and 6 possible colors (red, blue, green, orange, yellow, or gray). Introduced by Mittal et al. (2021), this task consists of three question types: (i) the properties of single objects (*Unary*), such as color and shape, (ii) the relationship between two objects (*Binary*), and (iii) the relationship among three objects (*Ternary*). An example of question type (i) is: "What is the shape of the red object?". An example of question type (ii) or (iii) is: "What is the shape of the object that is farthest from the blue object?".

We follow the experiment settings used in Mittal et al. (2021). To extract image features, we use a single CNN layer with a kernel size of 15 and a stride size of 15. As our baseline model, we use a 4-layered Linear Transformer with shared parameters. The experiments on the sort-of-CLEVR task are repeated 10 times with distinct seeds. For training, we utilize the Adam optimizer with a batch size of 64 and a learning rate of $1e^{-4}$ for 100 training epochs.

### B.4 WIKITEXT-103 TASK

WikiText-103 task (Merity et al., 2016) comprises lengthy corpora from Wikipedia articles and evaluates the model's understanding of the probability distribution over a sequence of words through perplexity. The training set consists of 28,475 articles, while the validation and test sets contain 60 articles each.

We follow the experimental settings described in Schlag et al. (2021) for the WikiText-103 task. The training data are partitioned into segments of $L$ words (back-propagation span). During the

evaluation, with a batch size of one, we compute perplexity using a sliding window with a length of $L$. The perplexity computation considers only the last position of segments, except for the first segment, where all positions are evaluated. For training, we use Adam optimizer with a batch size of 96, an initial learning rate of $2.5e^{-4}$, and a learning rate warmup step of 2,000 for 120 epochs.

In the experiment, we use a 16-layered Linear Transformer and a 16-layered Delta Network as our baseline models, following Schlag et al. (2021). To mitigate the additional computational overhead by the AID, we strategically integrate our AID module into the baseline models at intervals of 4 out of the 16 layers. Fig. 10 shows the experimental results for two cases: (1) where AID is applied to the first layer and (2) where AID is applied to the fourth layer.

## C  HYPER-PARAMETER SETTINGS OF AID MODULE

Tables 6, 7, and 8 show our module's hyper-parameter settings for each task. Each parameter is denoted as follows: $N_{\text{com}}$ as the number of components, $N_{\text{iters}}$ as the number of iterations, $N_{\text{inputs}}$ as the number of input features, $D_{\text{inputs}}$ as the dimension of input features, $D_{\text{com}}$ as the dimension of components, $D_{\text{MLP}_{\text{update}}}$ as the dimension of $\text{MLP}_{\text{update}}$'s two layers, $D_{\text{MLP}_{\text{final}}}$ as the dimension of $\text{MLP}_{\text{final}}$, $p_{\text{dropout}}$ as the probability of Dropout. In particular, we set different values as $N_{\text{com}}$ in encoding and decoding, in line with the TPR functions of the baseline model (Schlag & Schmidhuber, 2018; Schlag et al., 2020).

Also, the gray background represents the baseline models' hyper-parameters. Each parameter is denoted as follows: $D_{\text{entity}}$ as the dimension of the entity vector, $D_{\text{relation}}$ as the dimension of the relation vector, $N_{\text{read}}$ as the number of read heads, $D_{\text{LSTM}}$ as the dimension of LSTM's hidden state, $D_{\text{memory}}$ as the dimension of FWM's memory, $N_{\text{heads}}$ as the number of attention heads, $D_{\text{heads}}$ as the dimension of attention heads.

Table 6: Hyper-parameter settings of the AID on TPR-RNN.

|  | Systematic bAbI task |
|---|---|
| $D_{\text{entity}}$ | 90 |
| $D_{\text{relation}}$ | 20 |
| $N_{\text{com}}^{\text{enc}}$ | 5 |
| $N_{\text{com}}^{\text{dec}}$ | 4 |
| $N_{\text{iters}}$ | 2 |
| $D_{\text{inputs}}$ | 64 |
| $D_{\text{com}}$ | 64 |
| $D_{\text{MLP}_{\text{update}}}$ | (128, 64) |
| $D_{\text{MLP}_{\text{final}}}$ | 64 |
| $p_{\text{dropout}}$ | 0.5 |

Table 7: Hyper-parameter settings of the AID on FWM.

| | SAR task | Systematic bAbI task |
|---|---|---|
| $N_{\text{read}}$ | 1 | 3 |
| $D_{\text{LSTM}}$ | 256 | 256 |
| $D_{\text{memory}}$ | 32 | 32 |
| $N_{\text{com}}^{\text{enc}}$ | 3 | 3 |
| $N_{\text{com}}^{\text{dec}}$ | $1 + N_{\text{read}}$ | $1 + N_{\text{read}}$ |
| $N_{\text{iters}}$ | 2 | 3 |
| $N_{\text{inputs}}$ | 3 | 6 |
| $D_{\text{inputs}}$ | 32 | 32 |
| $D_{\text{com}}$ | 32 | 32 |
| $D_{\text{MLP}_{\text{update}}}$ | (64, 32) | (64, 32) |
| $D_{\text{MLP}_{\text{final}}}$ | 32 | 32 |
| $p_{\text{dropout}}$ | 0.5 | 0.5 |

Table 8: Hyper-parameter settings of the AID on Linear Transformer.

| | Sort-of-CLEVR task | WikiText-103 task |
|---|---|---|
| $N_{\text{heads}}$ | 4 | 8 |
| $D_{\text{heads}}$ | 64 | 16 |
| $N_{\text{com}}^{\text{enc}}$ | $2 * N_{\text{heads}}$ | $2 * N_{\text{heads}}$ |
| $N_{\text{iters}}$ | 2 | 2 |
| $N_{\text{inputs}}$ | $2 * N_{\text{heads}}$ | $2 * N_{\text{heads}}$ |
| $D_{\text{inputs}}$ | $D_{\text{heads}}$ | $D_{\text{heads}}$ |
| $D_{\text{com}}$ | $D_{\text{heads}}$ | $D_{\text{heads}}$ |
| $D_{\text{MLP}_{\text{update}}}$ | $(2 * D_{\text{heads}}, D_{\text{heads}})$ | $(2 * D_{\text{heads}}, D_{\text{heads}})$ |
| $D_{\text{MLP}_{\text{final}}}$ | $D_{\text{heads}}$ | $D_{\text{heads}}$ |
| $p_{\text{dropout}}$ | 0.5 | 0.5 |

## D    ABLATION STUDY

**Module**    In Fig. 7, we show the effect of incorporating a residual connection (He et al., 2016) to `query` and a concatenation after the iterative process. Each of these techniques demonstrates the enhancement in performance when applied to the *sys-bAbI* task. However, the concatenation technique exhibits unstable and lower performance on the SAR task. Remarkably, when these techniques are jointly applied to the AID, a clear improvement is evident across both tasks.

**Activation function**    In Fig. 8, we show the effect of activation functions at dot-product attention. ELU activation function (Clevert et al., 2015) contributes to improved stability and enhanced performance across both tasks.

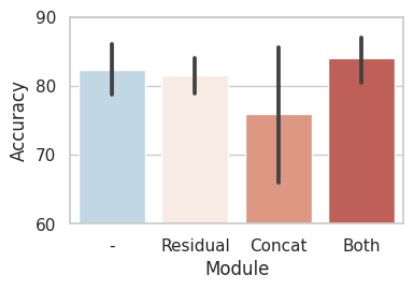
(a) Systematic Associative Recall task

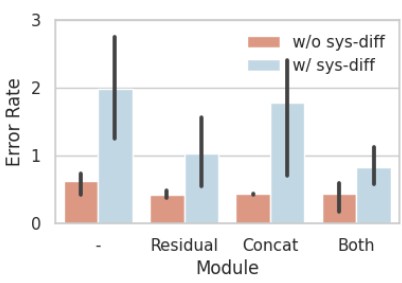
(b) Systematic bAbI task

Figure 7: Ablation Study for module.

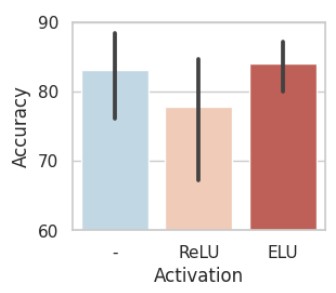
(a) Systematic Associative Recall task

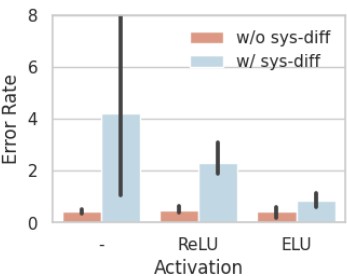
(b) Systematic bAbI task

Figure 8: Ablation Study for activation functions.

# E    ADDITIONAL EXPERIMENT RESULTS

## E.1    SYSTEMATIC BABI TASK

Table 9: Additional experiment results on *sys-bAbI* task for 3 seeds.

| $N_{\text{iters}}$ | $N_{\text{inputs}}$ | *w/o sys diff* | *w/ sys diff* | **Gap** | # params |
|---|---|---|---|---|---|
| 2 | 5 | $0.69 \pm 0.11$ | $3.69 \pm 3.21$ | 3.00 | 1.13 $M$ |
|   | 6 | $0.69 \pm 0.24$ | $3.28 \pm 2.27$ | 2.59 | 1.26 $M$ |
| 3 | 3 | $0.67 \pm 0.19$ | $2.80 \pm 0.88$ | 2.13 | 0.87 $M$ |
|   | 4 | $0.67 \pm 0.07$ | $3.42 \pm 0.57$ | 2.75 | 1.00 $M$ |
|   | 5 | $\mathbf{0.39 \pm 0.03}$ | $0.87 \pm 0.34$ | 0.48 | 1.13 $M$ |
|   | 6 | $0.43 \pm 0.18$ | $\mathbf{0.83 \pm 0.23}$ | **0.40** | 1.26 $M$ |
| 4 | 5 | $0.55 \pm 0.08$ | $2.73 \pm 1.30$ | 2.18 | 1.13 $M$ |
|   | 6 | $0.54 \pm 0.18$ | $3.32 \pm 0.80$ | 2.78 | 1.26 $M$ |

### E.2 WIKITEXT-103 TASK

Table 10: Additional experiment results on WikiText-103 task.

| Model | | Valid | Test | # params |
|---|---|---|---|---|
| Linear Transformer[†] | | 36.473 | 37.533 | 44.02 $M$ |
| | + AID (1st layer) | **36.159** (0.314 ↓) | **37.151** (0.382 ↓) | 44.16 $M$ |
| | + AID (4th layer)) | 36.301 (0.172 ↓) | 37.425 (0.108 ↓) | 44.16 $M$ |
| Delta Network[†] | | 35.640 | 36.659 | 44.03 $M$ |
| | + AID (1st layer)) | 35.361 (0.279 ↓) | **36.253** (0.406 ↓) | 44.18 $M$ |
| | + AID (4th layer)) | **35.355** (0.285 ↓) | 36.291 (0.368 ↓) | 44.18 $M$ |

## F COMPARISON TO BASELINE WITH MORE PARAMETERS

We conduct experiments with baseline models with more parameters. In the WikiText-103 task, we increase the size of the feed-forward network in the attention part of the baseline model. The size increase is applied to the exact same positions of the model where AID is adopted, for a fair comparison with our AID-assisted network architecture. In other tasks, we apply a different method to increase the model parameters, increasing either the hidden or head size of the baseline models. Fig. 9, Tables 11, 12, and 13 show that improvements obtained with AID do not merely come from the number of increased parameters in the models.

### F.1 SYSTEMATIC ASSOCIATIVE RECALL TASK

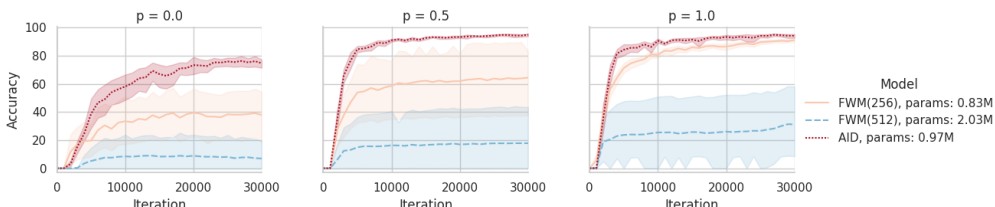

Figure 9: Comparison to baseline with more parameters on the SAR task for 10 seeds.

### F.2 SYSTEMATIC BABI TASK

Table 11: Comparison to baseline with more parameters on the *sys-bAbI* task on for 10 seeds.

| Model | $D_{\text{LSTM}}$ | w/o sys diff | w/ sys diff | Gap | # params |
|---|---|---|---|---|---|
| FWM | 256 | 0.79 ± 0.14 | 2.85 ± 1.61 | 2.35 | 0.73 $M$ |
| + AID | 256 | **0.45** ± 0.16 (0.34 ↓) | **1.21** ± 0.66 (1.64 ↓) | **0.76** (1.59 ↓) | 1.23 $M$ |
| | 512 | 0.75 ± 0.20 | 2.16 ± 1.33 | 1.41 | 1.89 $M$ |

## F.3 SORT-OF-CLEVR TASK

Table 12: Comparison to baseline with more parameters on the Sort-of-CLEVR task for 10 seeds.

| Model | $N_{\text{heads}}$ | $D_{\text{heads}}$ | Unary Accuracy | Binary Accuracy | Ternary Accuracy | # params |
|---|---|---|---|---|---|---|
| Linear Transformer | 8 | 32 | $82.5 \pm 18.3$ | $78.3 \pm 2.7$ | $60.0 \pm 5.0$ | $0.68\ M$ |
| + AID | | | $\textbf{98.9} \pm 0.2\ (16.4\ \uparrow)$ | $78.0 \pm 0.6\ (0.3\ \downarrow)$ | $61.0 \pm 1.7\ (1.0\ \uparrow)$ | $0.83\ M$ |
| Linear Transformer | 4 | 64 | $69.3 \pm 14.8$ | $75.5 \pm 1.3$ | $56.4 \pm 4.3$ | $0.68\ M$ |
| + AID | | | $\textbf{98.9} \pm 0.2\ (29.6\ \uparrow)$ | $\textbf{78.6} \pm 0.3\ (3.1\ \uparrow)$ | $\textbf{63.7} \pm 1.2\ (7.3\ \uparrow)$ | $0.83\ M$ |
| Linear Transformer | 8 | 64 | $57.5 \pm 5.6$ | $59.7 \pm 11.1$ | $53.2 \pm 1.4$ | $2.55\ M$ |
| Linear Transformer | 4 | 128 | $57.9 \pm 1.3$ | $59.9 \pm 4.7$ | $52.2 \pm 0.9$ | $2.55\ M$ |

## F.4 WIKITEXT-103 TASK

Table 13: Comparison to baseline with more parameters on the WikiText-103 task. $^{*}$ indicates an increasing feed-forward size of the attention layer.

| Model | Valid | Test | # params |
|---|---|---|---|
| Linear Transformer | 36.473 | 37.533 | $44.02\ M$ |
| + AID | $\textbf{36.159}\ (0.314\ \downarrow)$ | $\textbf{37.151}\ (0.382\ \downarrow)$ | $44.16\ M$ |
| Linear Transformer$^{*}$ | 36.452 | 37.306 | $44.22\ M$ |
| Delta Network | 35.640 | 36.659 | $44.03\ M$ |
| + AID | $\textbf{35.361}\ (0.279\ \downarrow)$ | $\textbf{36.253}\ (0.406\ \downarrow)$ | $44.18\ M$ |
| Delta Network$^{*}$ | 35.468 | 36.639 | $44.23\ M$ |

## G ADDITIONAL QUALITATIVE ANALYSIS

We conduct qualitative analysis for generated representations on the bAbI task. For this analysis, we consider the following two sentences where the desired answer is "*kitchen*".

- (*w/o sys-diff*) *sandra moved to the office. afterward she journeyed to the kitchen. daniel went to the hallway. then he journeyed to the kitchen. where is sandra?*
- (*w/ sys-diff*) *julie moved to the office. afterward she journeyed to the kitchen. bill went to the hallway. then he journeyed to the kitchen. where is julie?*

Figs. 10 and 11 show the similarity between *roles* across the input sequence. FWM and AID exhibit a high correlation when the sentence subjects are identical, suggesting that word-level TPR-based models might learn to represent symbolic structures sentence-by-sentence. Notably, FWM shows a high intra-sentence word correlation, while AID shows a high correlation at sentence terminations (indicated by "."). As highlighted in the yellow box comparison, FWM, when confronted with unfamiliar subjects (*w/ sys-diff* case), shows a decreased correlation between relevant sentences and an increased correlation among irrelevant ones. Conversely, AID maintains consistent results irrespective of systematic differences.

Furthermore, we explore similarity patterns between *roles* and *unbinding operators*, as done in Schlag et al. (2020). We utilize the *roles* generated at each time step of the input sequence and the *unbinding operators* generated at "?" for each of the read heads ($N_r = 3$). Figs. 12 and 13 reveal that both models exhibit a high correlation at the end of each sentence ("."). As seen from the yellow box, the FWM struggles to link FWM struggles to link the "." of the question-related sentences in the *sys-diff* case, which may explain the prediction of an incorrect answer ("*office*"). In contrast, the AID shows consistent patterns and accurately predicts the correct answer. These findings elucidate why FWM fails and AID succeeds in tackling the *sys-bAbI* task.

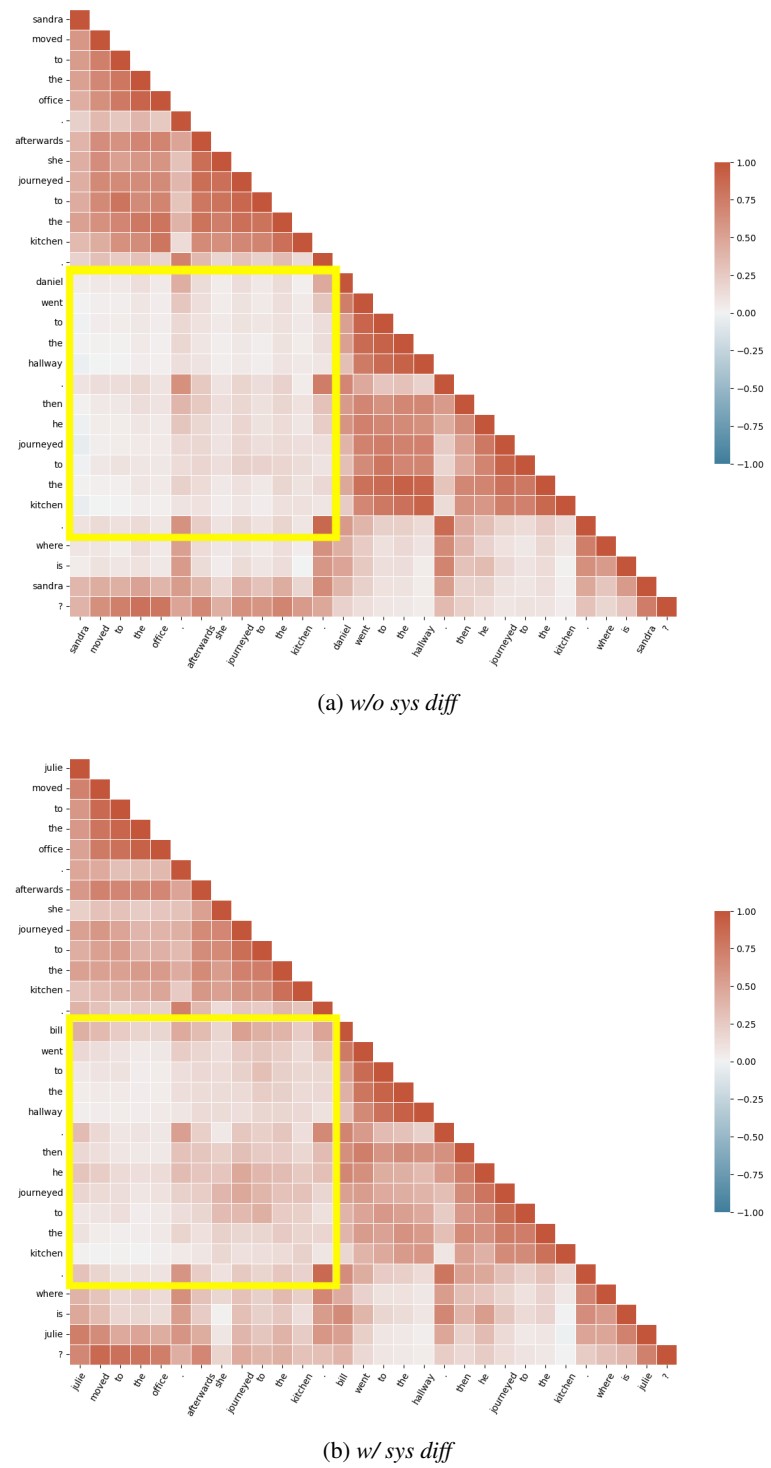

(a) *w/o sys diff*

(b) *w/ sys diff*

Figure 10: The heatmap of FWM for the cosine similarity between *roles* on the *sys-bAbI* task. We use the *roles* generated at each time step of the input sequence.

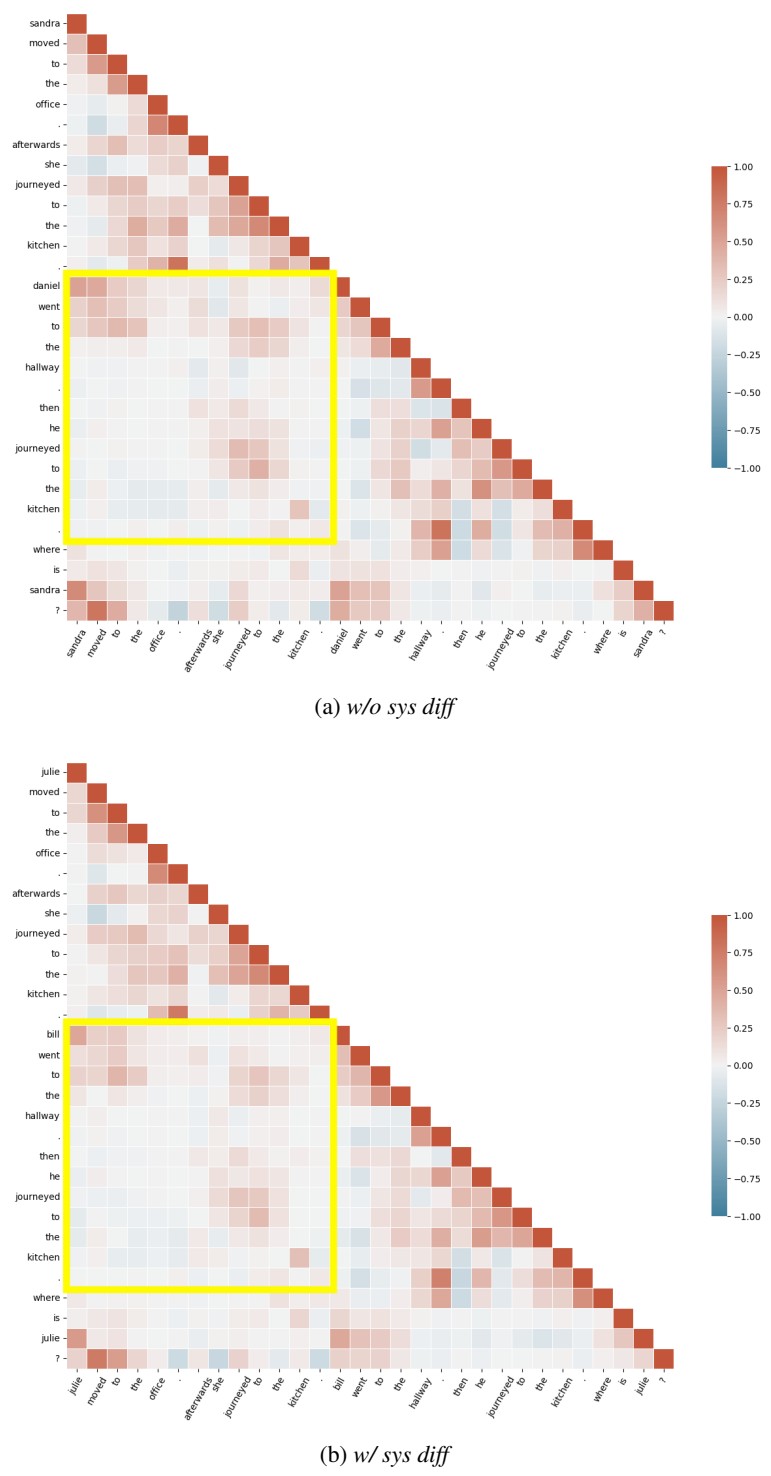

Figure 11: The heatmap of AID for the cosine similarity between *roles* on the *sys-bAbI* task. We use the *roles* generated at each time step of the input sequence.

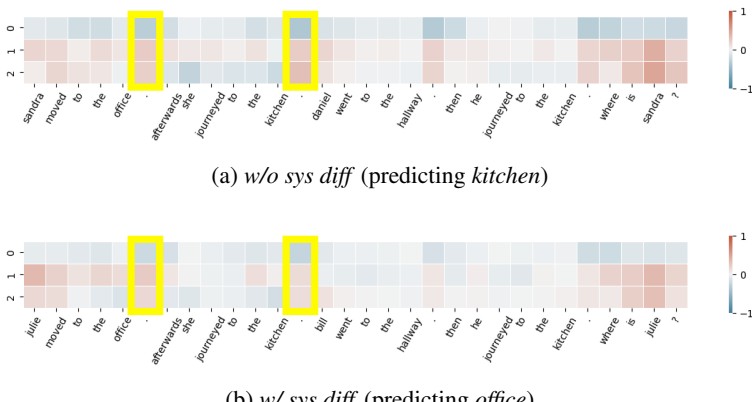

(a) *w/o sys diff* (predicting *kitchen*)

(b) *w/ sys diff* (predicting *office*)

Figure 12: The heatmap of FWM for the cosine similarity between *roles* (x-axis) and *unbinding operators* (y-axis) on the *sys-bAbI* task. We use the *roles* generated at each time step of the input sequence and the *unbinding operators* generated at "?" for each of the read heads ($N_r = 3$).

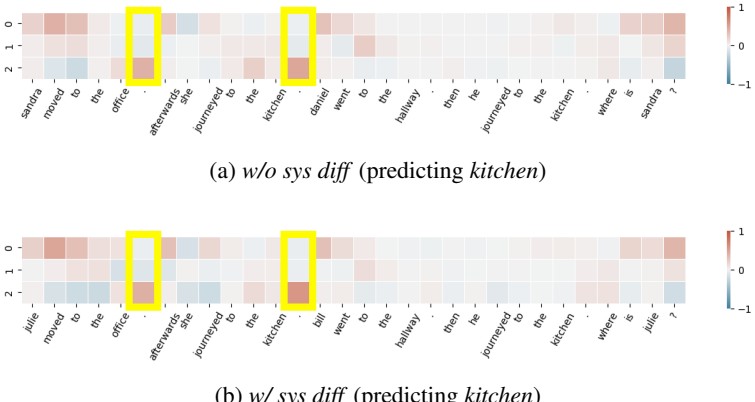

(a) *w/o sys diff* (predicting *kitchen*)

(b) *w/ sys diff* (predicting *kitchen*)

Figure 13: The heatmap of AID for the cosine similarity between *roles* (x-axis) and *unbinding operators* (y-axis) on the *sys-bAbI* task. We use the *roles* generated at each time step of the input sequence and the *unbinding operators* generated at "?" for each of the read heads ($N_r = 3$).

Table 14: The mean word error rate [%] in detail on the *sys-bAbI* task for 10 seeds.

| Task | TXL | | DAM | | STM | | TPR-RNN | | TPR-RNN$_{+AID}$ | | FWM | | FWM$_{+AID}$ | |
|---|---|---|---|---|---|---|---|---|---|---|---|---|---|---|
| | w/o | w/ | w/o | w/ | w/o | w/ | w/o | w/ | w/o | w/ | w/o | w/ | w/o | w/ |
| 1: one supporting fact | 0.00 | 2.31 | 0.00 | 0.62 | 0.00 | 0.12 | 0.00 | 2.85 | 0.00 | 0.05 | 0.00 | 2.18 | 0.00 | 0.09 |
| 2: two supporting facts | 11.44 | 24.42 | 0.82 | 3.78 | 0.56 | 1.79 | 0.36 | 5.91 | 0.19 | 2.73 | 1.08 | 2.81 | 0.94 | 1.70 |
| 3: three supporting facts | 23.50 | 35.68 | 2.88 | 7.27 | 3.43 | 7.66 | 2.33 | 10.05 | 1.91 | 5.85 | 5.88 | 9.88 | 2.16 | 3.34 |
| 6: yes/no questions | 0.01 | 0.03 | 0.08 | 0.70 | 0.05 | 0.15 | 0.05 | 0.30 | 0.03 | 0.09 | 0.05 | 0.09 | 0.07 | 0.06 |
| 7: counting | 1.80 | 5.71 | 1.34 | 6.09 | 1.19 | 12.85 | 1.32 | 20.46 | 1.23 | 15.82 | 1.34 | 5.57 | 1.13 | 5.33 |
| 8: lists/sets | 0.84 | 3.64 | 0.31 | 2.31 | 0.23 | 3.77 | 0.64 | 3.90 | 0.31 | 3.08 | 0.36 | 1.74 | 0.22 | 1.47 |
| 9: simple negation | 0.02 | 0.28 | 0.04 | 0.10 | 0.07 | 0.34 | 0.15 | 0.67 | 0.12 | 0.58 | 0.08 | 0.10 | 0.02 | 0.11 |
| 10: indefinite knowl. | 0.43 | 0.57 | 0.06 | 0.27 | 0.07 | 0.59 | 0.32 | 1.19 | 0.29 | 0.45 | 0.28 | 0.44 | 0.41 | 0.44 |
| 11: basic coreference | 0.00 | 18.10 | 0.00 | 22.45 | 0.06 | 17.90 | 0.66 | 28.65 | 0.43 | 25.05 | 0.00 | 8.13 | 0.00 | 0.06 |
| 12: conjunction | 0.00 | 4.14 | 0.01 | 8.76 | 0.06 | 2.55 | 1.14 | 13.21 | 1.07 | 4.45 | 0.01 | 3.29 | 0.00 | 0.67 |
| 13: compound coref. | 0.00 | 1.79 | 0.04 | 8.83 | 0.03 | 1.05 | 1.98 | 6.51 | 2.34 | 4.46 | 0.01 | 1.42 | 0.00 | 0.41 |
| 14: time reasoning | 6.46 | 7.95 | 0.13 | 1.76 | 0.20 | 1.48 | 0.52 | 11.19 | 0.36 | 4.66 | 0.86 | 2.47 | 0.43 | 0.79 |

Table 15: Word sets in the SAR task. We refer to the words in link

| Set | Word |
| --- | --- |
| $X_1$ | a, ability, able, about, above, accept, according, account, across, act, action, activity, actually, add, address, administration, admit, adult, affect, after, again, against, age, agency, agent, ago, agree, agreement, ahead, air, all, allow, almost, alone, along, already, also, although, always, american, among, amount, analysis, and, animal, another, answer, any, anyone, anything, appear, apply, approach, area, argue, arm, around, arrive, art, article, artist, as, ask, assume, at, attack, attention, attorney, audience, author, authority, available, avoid, away, baby, back, bad, bag, ball, bank, bar, base, be, beat, beautiful, because, become, bed, before, begin, behavior, behind, believe, benefit, best, better, between, beyond, big, bill, billion, bit, black, blood, blue, board, body, book, born, both, box, boy, break, bring, brother, budget, build, building, business, but, buy, by, call, camera, campaign, can, cancer, candidate, capital, car, card, care, career, carry, case, catch, cause, cell, center, central, century, certain, certainly, chair, challenge, chance, change, character, charge, check, child, choice, choose, church, citizen, city, civil, claim, class, clear, clearly, close, coach, cold, collection, college, color, come, commercial, common, community, company, compare, computer, concern, condition, conference, congress, consider, consumer, contain, continue, control, cost, could, country, couple, course, court, cover, create, crime, cultural, culture, cup, current, customer, cut, dark, data, daughter, day, dead, deal, death, debate, decade, decide, decision, deep, defense, degree, democrat, democratic, describe, design, despite, detail, determine, develop, development, die, difference, different, difficult, dinner, direction, director, discover, discuss, discussion, disease, do, doctor, dog, door, down, draw, dream, drive, drop, drug, during, each, early, east, easy, eat, economic, economy |
| $X_2 \cup X_3$ | edge, education, effect, effort, eight, either, election, else, employee, end, energy, enjoy, enough, enter, entire, environment, environmental, especially, establish, even, evening, event, ever, every, everybody, everyone, everything, evidence, exactly, example, executive, exist, expect, experience, expert, explain, eye, face, fact, factor, fail, fall, family, far, fast, father, fear, federal, feel, feeling, few, field, fight, figure, fill, film, final, finally, financial, find, fine, finger, finish, fire, firm, first, fish, five, floor, fly, focus, follow, food, foot, for, force, foreign, forget, form, former, forward, four, free, friend, from, front, full, fund, future, game, garden, gas, general, generation, get, girl, give, glass, go, goal, good, government, great, green, ground, group, grow, growth, guess, gun, guy, hair, half, hand, hang, happen, happy, hard, have, he, head, health, hear, heart, heat, heavy, help, her, here, herself, high, him, himself, his, history, hit, hold, home, hope, hospital, hot, hotel, hour, house, how, however, huge, human, hundred, husband, i, idea, identify, if, image, imagine, impact, important, improve, in, include, including, increase, indeed, indicate, individual, industry, information, inside, instead, institution, interest, interesting, international, interview, into, investment, involve, issue, it, item, its, itself, job, join, just, keep, key, kid, kill, kind, kitchen, know, knowledge, land, language, large, last, late, later, laugh, law, lawyer, lay, lead, leader, learn, least, leave, left, leg, legal, less, let, letter, level, lie, life, light, like, likely, line, list, listen, little, live, local, long, look, lose, loss, lot, love, low, machine, magazine, main, maintain, major, majority, make, man, manage, management, manager, many, market, marriage, material, matter |
| $Y_1$ | may, maybe, me, mean, measure, media, medical, meet, meeting, member, memory, mention, message, method, middle, might, military, million, mind, minute, miss, mission, model, modern, moment, money, month, more, morning, most, mother, mouth, move, movement, movie, mr, mrs, much, music, must, my, myself, name, nation, national, natural, nature, near, nearly, necessary, need, network, never, new, news, newspaper, next, nice, night, no, none, nor, north, not, note, nothing, notice, now, nt, number, occur, of, off, offer, office, officer, official, often, oh, oil, ok, old, on, once, one, only, onto, open, operation, opportunity, option, or, order, organization, other, others, our, out, outside, over, own, owner, page, pain, painting, paper, parent, part, participant, particular, particularly, partner, party, pass, past, patient, pattern, pay, peace, people, per, perform, performance, perhaps, period, person, personal, phone, physical, pick, picture, piece, place, plan, plant, play, player, pm, point, police, policy, political, politics, poor, popular, population, position, positive, possible, power, practice, prepare, present, president, pressure, pretty, prevent, price, private, probably, problem, process, produce, product, production, professional, professor, program, project, property, protect, prove, provide, public, pull, purpose, push, put, quality, question, quickly, quite, race, radio, raise, range, rate, rather, reach, read, ready, real, reality, realize, really, reason, receive, recent, recently, recognize, record, red, reduce, reflect, region, relate, relationship, religious, remain, remember, remove, report, represent, republican, require, research, resource, respond, response, responsibility, rest, result, return, reveal, rich, right, rise, risk, road, rock, role, room, rule, run, safe, same, save, say, scene, school, science, scientist, score, sea, season, seat, second, section, security, see |
| $Y_2$ | seek, seem, sell, send, senior, sense, series, serious, serve, service, set, seven, several, sex, sexual, shake, share, she, shoot, short, shot, should, shoulder, show, side, sign, significant, similar, simple, simply, since, sing, single, sister, sit, site, situation, six, size, skill, skin, small, smile, so, social, society, soldier, some, somebody, someone, something, sometimes, son, song, soon, sort, sound, source, south, southern, space, speak, special, specific, speech, spend, sport, spring, staff, stage, stand, standard, star, start, state, statement, station, stay, step, still, stock, stop, store, story, strategy, street, strong, structure, student, study, stuff, style, subject, success, successful, such, suddenly, suffer, suggest, summer, support, sure, surface, system, table, take, talk, task, tax, teach, teacher, team, technology, television, tell, ten, tend, term, test, than, thank, that, the, their, them, themselves, then, theory, there, these, they, thing, think, third, this, those, though, thought, thousand, threat, three, through, throughout, throw, thus, time, to, today, together, tonight, too, top, total, tough, toward, town, trade, traditional, training, travel, treat, treatment, tree, trial, trip, trouble, true, truth, try, turn, tv, two, type, under, understand, unit, until, up, upon, us, use, usually, value, various, very, victim, view, violence, visit, voice, vote, wait, walk, wall, want, war, watch, water, way, we, weapon, wear, week, weight, well, west, western, what, whatever, when, where, whether, which, while, white, who, whole, whom, whose, why, wide, wife, will, win, wind, window, wish, with, within, without, woman, wonder, word, work, worker, world, worry, would, write, writer, wrong, yard, yeah, year, yes, yet, you, young, your, yourself |

