# OpenReview forum: "Attention-based Iterative Decomposition for Tensor Product Representation"
_ICLR.cc/2024/Conference — ICLR 2024 poster_

### Official Review · Reviewer_Qfs2 · 2023-11-01

**Soundness:** 3 good
**Presentation:** 3 good
**Contribution:** 3 good
**Rating:** 8
**Confidence:** 2

**Summary:**

The authors try to apply the Attention mechanism in the tensor product representation models. They also showed that the proposed AID block can be easily incorporated into many existing networks. Experiments show the advantages of introducing the AID block in previous network architectures.

**Strengths:**

1. The authors proposed a new Attention based module for TPR. The proposed module can be combined with existing structures such as TPR-RNN, FWM and Linear Transformers.
2. The authors conducted extensive experiments including ablation studies to show the advantages of the AID module and influences of hyperparameters.
3. Code for all experiments is provided.

**Weaknesses:**

The authors mentioned that one advantage of TPR is to represent symbolic structures. I am wondering if this was demonstrated in experiments. I am not familiar with these tasks, but I did not find descriptions about this issue in experiments.

**Questions:**

How is the scalability and complexity of the proposed AID module?

---

> ### Author Response · Authors · 2023-11-19
>
> We thank the reviewer for their helpful comments. Also, we updated the manuscript to reflect review comments and marked revision with red color. Our responses to your concerns are as follows:
>
> > The authors mentioned that one advantage of TPR is to represent symbolic structures. I am wondering if this was demonstrated in experiments. I am not familiar with these tasks, but I did not find descriptions about this issue in experiments.
>
> To address your inquiry, we first would like to explain TPR. TPR is a general method that represents the symbolic structure of data in vector spaces. Also, it provides symbolic information through the decoding process. To gain more insight into how TPR represents the symbolic properties of data, let us consider a scenario where two objects (a red rectangle and a blue circle) are presented in a room, and our objective is to determine the circle’s color. Initially, the two objects are encoded into TPR form by superimposing embedding representations for each object, expressed as $T= d_{\text{red}} \otimes d_{\text{rectangle}} + d_{\text{blue}} \otimes d_{\text{circle}}$. Afterward, it decodes the color information of the circle from TPR with an inner product, $T \cdot d_{\text{circle}} = d_{\text{blue}}$. In this example, object colors are *fillers* and object shapes are *roles* and the *unbinding operators* in the TPR components.
>
> Because of those TPR’s characteristics, the assessment of how accurately the models represent the symbolic structure in the TPR form hinges on their capacity to generate appropriate *role* and *filler* representations. Moreover, the specific task requirements determine these *roles* and *fillers*. To investigate the ability of models to generate appropriately structured representations, we designed a synthetic task named the Systematic Associative Recall task. This task entails the clear mapping of generative factors to specific TPR elements, such as associating $x$ with *role* and $y$ with *filler*. Our quantitative and qualitative analyses in Section 4.3.1 (Originally, Section 3.3.1) reveal that the AID generates structural representations that better conform to TPR conditions than the baseline model.
>
>
> ***
>
> > How is the scalability and complexity of the proposed AID module?
>
> The iterative attention mechanism of the AID enables it to scale well with the number of TPR components. This scalability allows the AID to be adapted to any TPR-based model. However, an increment in the number of components potentially impacts our routing strategy (learning of initial components), closely related to the training stability. Also, since the AID module demands additional computational complexity $\mathcal{O}( N_\text{input} N_\text{com} D_\text{com} )$ per iteration at each time step, this increment results in a slowdown of the overall model operations.

---

> > ### Comment · Reviewer_Qfs2 · 2023-11-23
> >
> > Thanks for the authors’ response. I am not familiar with this field. However, this paper presents some interesting applications of TPR and is well written. The experimental results seem valid. Therefore, I will keep my rating and a low confidence level.

---

> > > ### Author Response · Authors · 2023-11-23
> > > **Thank you for your time**
> > >
> > > We thank Reviewer Qfs2 for your time and your positive feedback on the rebuttal.

---

### Official Review · Reviewer_bYLE · 2023-11-01

**Soundness:** 3 good
**Presentation:** 4 excellent
**Contribution:** 3 good
**Rating:** 6
**Confidence:** 3

**Summary:**

This paper proposes an Attention-based Iterative Decomposition (AID) module that uses a competitive attention mechanism to decompose sequential input features into structured representations (roles, fillers, and unbinding operators) to improve systematic generalization for Tensor Product Representation (TPR) based models.
The AID module is flexible enough to integrate with existing TPR-based models such as TPR-RNN, Fast Weight Memory, and Linear Transformer.
The experiments support the improvements, show AID produces more compositional and well-bound structural representations, and exemplify applications with large-scale real-world data.

**Strengths:**

- It is important to decompose sequential input to structured representations for systematic generalization, and the AID module enhances the performances for TPR-based models.

- The module design is simple and clean, so it may be expected to keep the advantage in general cases.

- It integrates with a wide range of TPR-based models in flexible ways.

**Weaknesses:**

(1) The WikiText-103 task shows the AID module performs well in a large-vocabulary language modeling task, but it seems not to be a systematic generalization task.

**Questions:**

(2) Do the intermediate TPR components always keep TPR conditions (the three key conditions required by TPR)?
For example, in integrating with TPR-RNN, the input features to the AID module $x_t$ are a set of word vectors, which may be in any form.
Does the AID module convert the input features to TPR?

(3) TPR has its properties, such as the separation of roles and fillers.
Does the AID module use TPR properties in the module design, e.g., use role for attention key?

(4) Though the AID module is designed to enhance TPR-based models, is it also informative to compare it with non-TPR-based models in experiments?

---

> ### Author Response · Authors · 2023-11-19
>
> We thank the reviewer for their helpful comments. Also, we updated the manuscript to reflect review comments and marked revision with red color. Our responses to your concerns are as follows:
>
> > (1) The WikiText-103 task shows the AID module performs well in a large-vocabulary language modeling task, but it seems not to be a systematic generalization task.
>
> We think our description of the purpose of the experiment on the WikiText-103 may not be good enough. We updated the WikiText-103 part of the manuscript for a better illustration of our intention. [First paragraph of Section 4.4 (originally, Section 3.4)]
>
> Our AID achieves larger improvements on various systematic generalization tasks. In addition to these improvements, we extend our evaluation to assess the effectiveness of the AID on a more practical task, the WikiText-103 task, which may not be explicitly designed to evaluate the systematic generalization capability but is a fundamental problem. In this task, we aim to show the effectiveness of the AID on TPR-based models even when the task is not for systematic generalization. The WikiText-103 results show the potential of the AID for enhancing performance even on large-scale real tasks.
>
> ***
>
> > (2) Do the intermediate TPR components always keep TPR conditions (the three key conditions required by TPR)? For example, in integrating with TPR-RNN, the input features to the AID module are a set of word vectors, which may be in any form. Does the AID module convert the input features to TPR?
>
> (*Do the intermediate TPR components always keep TPR conditions?*) **No, the intermediate TPR components do not always keep TPR conditions.** Given the relationship between accuracy and adherence to TPR conditions, we believe Fig. 5(c) indirectly addresses your inquiry. This case is akin to making predictions based on the intermediate TPR components. Notably, when the number of attention iterations is limited, the outcomes demonstrate lower performance, indicating a potential failure to adhere to TPR conditions.
>
> (*Does the AID module convert the input features to TPR?*) **Yes.** More precisely, the AID module indeed plays a role in converting the input features into structured representations that serve as the constituents of TPR, such as *role* and *filler*.
>
>
>
> ***
>
> > (3) TPR has its properties, such as the separation of roles and fillers. Does the AID module use TPR properties in the module design, e.g., use role for attention key?
>
> **Our AID maintains the separation of *roles* and *fillers*, a key TPR property, in the module design.** As you mentioned, TPR operates by explicitly separating the information at the representation level into distinct symbols. The AID links each symbolic meaning to different slots during iterative attention to maintain this representational separation. These slots compete while being updated individually and independently throughout the competitive attention process.
>
>
> ***
>
> > (4) Though the AID module is designed to enhance TPR-based models, is it also informative to compare it with non-TPR-based models in experiments?
>
> Our purpose for the comparison to non-TPR-based models is to measure the systematic generalization capability of TPR-based models. We compare TPR-based memory networks (TPR-RNN and FWM) to state-of-the-art memory networks in the bAbI task. The experimental results indicate that TPR-based models show better systematic generalization performance than others. Furthermore, combined with those TPR-based models, our AID not only enhances the systematic generalization capability but also achieves state-of-the-art results.

---

> > ### Comment · Reviewer_bYLE · 2023-11-19
> >
> > Thank you for the answers and updates. I like to keep the score.

---

> > > ### Author Response · Authors · 2023-11-23
> > > **Thank you for your time**
> > >
> > > We thank Reviewer bYLE for your time and your positive feedback on the rebuttal.

---

### Official Review · Reviewer_bmNb · 2023-11-03

**Soundness:** 3 good
**Presentation:** 3 good
**Contribution:** 2 fair
**Rating:** 6
**Confidence:** 2

**Summary:**

This paper aims to improve Tensor Product Representation (TPR) for systematic generalization tasks. The authors propose an Attention-based Iterative Decomposition (AID) module, which is plug-and-play and can be easily integrated into existing TPR models. AID is conceptually similar to Slot Attention, but with special designs tailored towards the task. Experimental results show that AID consistently improves existing TPR methods across a broad range of tasks.

**Strengths:**

- The considered challenge, roles/fillers decomposition, is indeed very similar to the object binding problem in object-centric learning (OCL). Therefore, it is intuitive to apply the SOTA OCL module Slot Attention here.
- The experimental evaluations are thorough. AID shows consistent and non-marginal improvement in all the tasks.
- The ablation and adapted designs from the original Slot Attention are insightful.

**Weaknesses:**

My background is in OCL so I am unfamiliar with these tasks and baselines. One concern I have is all the tasks (except the WikiText-103 one) are very simple. I understand that areas in the early stage experiment on simple data. However, for example for the CLEVR VQA task, people can train a Slot Attention model to extract object-centric features, and then attach a small Transformer head to predict the question's answer. According to my own experience, such a naive baseline can already achieve nearly perfect accuracy (on the original CLEVR dataset, not Sort-of-CLEVR). Therefore, it is hard for me to assess the importance of this paper.

Also, what is the difference in model size and computation cost of baselines with and without AID? For example, on the WikiText-103 task, the authors mention that they do not insert AID in every layer due to computation concerns. I wonder how will the baselines perform if they have more parameters.

**Questions:**

The Orthogonality Analysis in Sec. 3.1.1 shows that AID also helps extract more orthogonal *roles*. I am curious why this is the case. In my own experience with Slot Attention, the object-centric features (slots) are usually entangled, as there is no loss to force them to be orthogonal. Any insights here?

---

> ### Author Response · Authors · 2023-11-19
>
> We thank the reviewer for their helpful comments. Also, we updated the manuscript to reflect review comments and marked revision with red color. Our responses to your concerns are as follows:
>
> > My background is in OCL so I am unfamiliar with these tasks and baselines. One concern I have is all the tasks (except the WikiText-103 one) are very simple … Therefore, it is hard for me to assess the importance of this paper.
>
> In response to your inquiry, we highlight that our AID is a general drop-in module that can be adapted to any TPR-based model and enhances its systematic generalization capability in various domains. The AID seems similar to Slot Attention in certain aspects because both employ competitive attention. However, even though Slot Attention can be a solution to solve the CLEVR task, it cannot directly tackle the SAR task, the bAbI task, and the WikiText-103 task. This is likely because Slot Attention is designed to extract visual object representations. In contrast, our AID generates structured representations regardless of the domain. Combined with TPR-based models, it can solve various systematic generalization tasks. In the experiment, we show the effectiveness of the AID to enhance the systematic generalization capability of TPR-based models.
>
> ***
>
> > Also, what is the difference in model size and computation cost of baselines with and without AID? For example, on the WikiText-103 task, the authors mention that they do not insert AID in every layer due to computation concerns. I wonder how will the baselines perform if they have more parameters.
>
> We have detailed the difference in parameter counts and have included our findings with a comparison to more parameterized baselines in Appendix E.
>
> In response, we conducted experiments with a more parameterized baseline on all the tasks. In the WikiText-103 task, on the advice of Reviewer 8A4n, we increased the size of the feed-forward network in the attention part of the baseline model. The size increase is applied to the exact same positions of the model where AID is adopted, for a fair comparison with our AID-assisted network architecture. In other tasks, we adopted a different methodology, increasing either the hidden or head size of the baseline models. In the Table below, the experimental results indicate that our improvements do not merely come from the number of increased parameters in the models.
>
> | **WikiText-103 task** | Valid | Test | # params |
> | --- | --- | --- | --- |
> | Linear Transfomer | 36.473 | 37.533 | 44.02M |
> | Linear Transfomer (more params) | 36.452 | 37.306 | 44.22M |
> | Linear Transfomer (+ AID) | **36.159** | **37.151** | 44.16M |
> | Delta Network | 35.640 | 36.659 | 44.03M |
> | Delta Network (more params) | 35.468 | 36.639 | 44.23M |
> | Delta Network (+ AID) | **35.361** | **36.253** | 44.18M |
>
>
> | **bAbI task** | $D_\text{LSTM}$ | *w/o sys diff* | *w/ sys diff* | Gap | # params |
> | --- | --- | --- | --- | --- | --- |
> | FWM | 256 | 0.79 | 2.85 | 2.35 | 0.73 M |
> | FWM (more params) | 512 | 0.75 | 2.16 | 1.41 | 1.89 M |
> | FWM (+ AID) | 256 | **0.45** | **1.21** | **0.76** | 1.23 M |
>
>
> | **Sort-of-CLEVR task** | $N_\text{heads}$ | $D_\text{heads}$ | *Unary* | *Binary* | *Ternary* | # params |
> | --- | --- | --- | --- | --- | --- | --- |
> | Linear Transformer | 8 | 32 | 82.5 | 78.3 | 60.0 | 0.68 M |
> | Linear Transformer (+ AID) | 8 | 32 | **98.9** | 78.0 | 61.0 | 0.83 M |
> | --- | --- | --- | --- | --- | --- | --- |
> | Linear Transformer | 4 | 64 | 69.3 | 75.5 | 56.4 | 0.68 M |
> | Linear Transformer (+ AID) | 4 | 64 | **98.9** | **78.6** | **63.7** | 0.83 M |
> | --- | --- | --- | --- | --- | --- | --- |
> | Linear Transformer | 8 | 64 | 57.5 | 59.7 | 53.2 | 2.55 M |
> | --- | --- | --- | --- | --- | --- | --- |
> | Linear Transformer | 4 | 128 | 57.9 | 59.9 | 52.2 | 2.55 M |

---

> ### Author Response · Authors · 2023-11-19
>
> > The Orthogonality Analysis in Sec. 3.1.1 shows that AID also helps extract more orthogonal *roles*. I am curious why this is the case. In my own experience with Slot Attention, the object-centric features (slots) are usually entangled, as there is no loss to force them to be orthogonal. Any insights here?
>
> **The orthogonality is closely associated with the properties of TPR.** TPR operates by explicitly separating the information at the representation level into distinct symbols, such as *role* and *filler*. It is constituted by the tensor product of *roles* vectors and *fillers* vectors to represent the symbolic structure of data. In the decoding phase, the *roles* act as keys, facilitating the retrieval of associated *fillers* through inner product operations. Because of these TPR operations, TPR-based models should generate orthogonal *roles* to accurately recall *filler* information from their TPR-formed connectionist representation. During training, those models learn these TPR’s properties to perform correct TPR operations in a supervised manner for solving tasks. Our AID is structured to learn the compositional nature of data and generates structural representations that better conform to TPR conditions than the baseline model.

---

> > ### Comment · Reviewer_bmNb · 2023-11-22
> > **Re: Rebuttal**
> >
> > I thank the authors for the rebuttal and additional experiments. However, I still do not see a clear difference on the technical side compared to Slot Attention. I will maintain my current rating of 5. Since I am unfamiliar with the field, I have downgraded my confidence score and let other reviewers decide.

---

> > > ### Author Response · Authors · 2023-11-23
> > > **Thank you for your time**
> > >
> > > We thank Reviewer bmNb for your time and your constructive feedback on the rebuttal.

---

### Official Review · Reviewer_rEt9 · 2023-11-09

**Soundness:** 3 good
**Presentation:** 3 good
**Contribution:** 2 fair
**Rating:** 6
**Confidence:** 4

**Summary:**

The work proposes using iterative attention for learning Tensor Product Representations (TPR), meant to improve their systematic generalization capability, as measured through experiments over textual and visual reasoning tasks.

**Update following rebuttal:**

Dear authors, thank you for addressing the comments in my review.

I appreciate the additional experimental results over the bAbI task, both the analysis per question type on table 14 and the new attention maps -- these look great. Expanding the discussion in the related work section about how the approach compares to slot attention is useful too. Finally, I appreciate the detailed response to my review and to those of other reviewers.

While I agree with reviewer bmNb that novelty-wise, the technical difference between slot attention and the paper is not large, nevertheless, for the usefulness of applying it in the context of TPRs and, most importantly, for the thoroughness of the rebuttal and paper updates, including performing additional experiments following the reviewers' comment, I'm happy to raise my score.

**Strengths:**

- **Idea**: TPRs and attention fit well together: identifying and extracting the role and filler components seems like a natural application of attention and so the integration between them makes a lot of sense to me.
- **Evaluation**: Experiments are conducted on multiple datasets including both textual and visual modalities as well as both synthetic and realistic data (bAbI, Sort-of-CLEVR, WikiText and the Systematic Associative Recall (SAR) task). The experiments investigate using the attention module to extend several related models (TPR-RNNs, Fast Weight Memory, and Linear Transformers). Both quantitative (through e.g. DCI, downstream performance) and a bit of qualitative analysis (visualization of similarity between the representations of the TPR components). Overall these support the approach’s flexibility.
- **Clarity**: The presentation is good and the paper is clearly written and well-organized. The introduction and model sections do a good job motivating the idea and presenting the necessary background and preliminaries. The overview figure is very helpful. Detailed description is provided for each of the 3 inspected models and the 4 tasks. The supplementary is also good, providing implementation details and ablation experiments.

**Weaknesses:**

- **Novelty**: The iterative attention decomposition works very similarly to slot attention, reducing the technical contribution of the paper. The paper introduces the idea as a novel attention-based module, not making it clear enough that effectively this strongly relies on slot attention. The comparison to slot attention appears only at the very end of the paper. Since the approach integrates together existing ideas, it will make sense in this case that the related work section will appear earlier on, before the model section.
- **Empirical Results**: The improvements for WikiText (perplexity) and disentanglement (DCI) are relatively low. On the other hand, we see larger improvements on bAbI and Sort-of-CLEVR.
- **Related Works**:  A more detailed comparison to the prior related works, in particular to “Enriching Transformers with Structured Tensor-Product Representations for Abstractive Summarization” that also integrates attention and TPRs. It is cited by the paper but more discussions on similarities and differences would be helpful.

**Questions:**

- **Qualitative Evaluation**: It would be particularly useful for this work to have more qualitative evaluation for both bAbI and sort-of-CLEVR. What do the different TPR components actually attend to? Does their behavior make sense over specific instances? What mistakes do they tend to make? What type of mistakes are made by the baselines and eliminated by the new approach? How do they behave over examples with unseen names (systematic generalization cases)? This type of analysis can significantly help in demonstrating the actual impact of integrating attention into TPRs, beyond the overall accuracy metrics.  For bAbI, a more detailed breakdown of the performance by question type or story length will also be helpful.

---

> ### Author Response · Authors · 2023-11-19
>
> We thank the reviewer for their helpful comments. Also, we updated the manuscript to reflect review comments and marked revision with red color. Our responses to your concerns are as follows:
>
> > **Novelty**: The iterative attention decomposition works very similarly to slot attention, reducing the technical contribution of the paper. The paper introduces the idea as a novel attention-based module, not making it clear enough that effectively this strongly relies on slot attention. The comparison to slot attention appears only at the very end of the paper. Since the approach integrates together existing ideas, it will make sense in this case that the related work section will appear earlier on, before the model section.
>
> Thanks for your suggestion. We have relocated the related work section to precede the method section. Also, we revised the second section of the related work to highlight our major distinction compared to the Slot Attention. [Second paragraph of Section 2 (originally, Section 4)]
>
> In certain aspects, AID seems similar to the Slot Attention method, because it employs the competitive attention for *role/filler* decomposition. However, in contrast to the original iterative attention which assumes the permutation-invariant slots, in our TPR decomposition problem, each slot representation cannot be equally linked to elements in TPR functions (e.g., should identify the appropriate slot for *filler*). Therefore, we introduced a trainable **Routing** ****mechanism to our iterative attention method to correctly integrate slot-based attention with the TPR framework. Our method enables the network to systematically associate each initial component to a specific structural component of TPR. **Through this Routing strategy, the AID can successfully generate the structural representations for TPR with iterative attention.**
>
> **Our AID is a drop-in module specifically designed to enhance the systematic generalization capability of the existing TPR-based models.** Also, we introduce seamless integration of the AID with various types of TPR-based models. Combined with these models, the AID shows effective generalization performance improvements.
>
>
>
> ***
>
> > **Empirical Results**: The improvements for WikiText (perplexity) and disentanglement (DCI) are relatively low. On the other hand, we see larger improvements on bAbI and Sort-of-CLEVR.
>
> We think our description of the purpose of the experiment on the WikiText-103 may not be good enough. We updated the WikiText-103 part of the manuscript for a better illustration of our intention. [First paragraph of Section 4.4 (originally, Section 3.4)]
>
> As you mentioned, our experiments show larger improvements on the SAR task, the bAbI task, and the Sort-of-CLEVR task that demand clearly systematic generalization capability from models. Our findings indicate that the AID substantially helps TPR-based models enhance their systematic generalization. In addition to these improvements, we extend our evaluation to assess the effectiveness of the AID on a more practical task, the WikiText-103 task, that may not explicitly demand systematic generalization capability but is an important problem. In this task, we aim to show the effectiveness of the AID on TPR-based models beyond systematic generalization tasks rather than aiming at achieving better performance with a larger gap. The WikiText-103 results show the potential of the AID for enhancing performance even on large-scale tasks.
>
> As other reviewers pointed out, one concern might be whether the improvement is just from more parameters. We conducted experiments with a more parameterized baseline on the WikiText-103 task to examine this. In the Table below, the experimental results indicate that our improvements are not merely attributable to increased parameters.
>
>
> | **WikiText-103 task** | Valid | Test | # params |
> | --- | --- | --- | --- |
> | Linear Transfomer | 36.473 | 37.533 | 44.02M |
> | Linear Transfomer (more params) | 36.452 | 37.306 | 44.22M |
> | Linear Transfomer (+ AID) | **36.159** | **37.151** | 44.16M |
> | Delta Network | 35.640 | 36.659 | 44.03M |
> | Delta Network (more params) | 35.468 | 36.639 | 44.23M |
> | Delta Network (+ AID) | **35.361** | **36.253** | 44.18M |
>
> The DCI framework provides quantitative metrics to evaluate the relationship between generative factors and representations. Despite the relatively lower improvement in DCI results, the AID generates more disentangled representations than the baseline model. One plausible explanation for this observation could be the task complexity. While the SAR task is relatively simple in disentangling individual items into representations, it further demands that the generated representations satisfy the TPR conditions to solve the task, which poses a challenge for the baseline model.

---

> ### Author Response · Authors · 2023-11-19
>
> > Related Works: A more detailed comparison to the prior related works, in particular to “Enriching Transformers with Structured Tensor-Product Representations for Abstractive Summarization” that also integrates attention and TPRs. It is cited by the paper but more discussions on similarities and differences would be helpful.
>
> Thanks for your feedback. In response, we have expanded the discussion within the related work section. [Second paragraph of Section 2 (originally, Section 4)] As you pointed out, our work has similarities with [1] in that both explore the integration between attention and TPRs. However, [1] is specifically designed for an abstractive summarization and relies on a pre-defined *role* embedding dictionary. Compared to [1], **the AID is a task-independent drop-in module that can be adapted to any TPR-based model. Also, it is designed to address a more fundamental problem, the decomposition of data into the appropriate *role* and *filler* simultaneously.**
>
> [1] Jiang, Y., Celikyilmaz, A., Smolensky, P., Soulos, P., Rao, S., Palangi, H., ... & Gao, J. (2021). Enriching transformers with structured tensor-product representations for abstractive summarization. *arXiv preprint arXiv:2106.01317*.
>
>
> ***
>
> > Qualitative Evaluation: It would be particularly useful for this work to have more qualitative evaluation for both bAbI and sort-of-CLEVR … This type of analysis can significantly help in demonstrating the actual impact of integrating attention into TPRs, beyond the overall accuracy metrics.
>
> Thanks for the constructive feedback. In response to your query, **we performed orthogonal analysis on the bAbI task**, as did in the SAR task. We have included the experimental results (including Figs. 10, 11, 12, and 13) in Appendix F. Our findings indicate that the AID shows consistent correlation patterns regardless of a systematic difference between data while the baseline shows an undesired pattern (e.g., a decreased correlation for question-relevant words) when processing unseen names. These findings explain why FWM fails and AID succeeds in tackling the *sys-bAbI* task.
>
> More specifically, we consider the following two sentences where the desired answer is "kitchen" for the orthogonal analysis.
>
> - (*w/o sys-diff*) sandra moved to the office. afterward she journeyed to the kitchen. daniel went to the hallway. then he journeyed to the kitchen. where is sandra?
>
> - (*w/ sys-diff*) julie moved to the office. afterward she journeyed to the kitchen. bill went to the hallway. then he journeyed to the kitchen. where is julie?
>
> Figs. 10 and 11 show the similarity between *roles* across the input sequence. FWM and AID exhibit a **high correlation when the sentence subjects are identical**, ****suggesting that word-level TPR-based models might learn to represent symbolic structures sentence-by-sentence. Notably, FWM shows a high intra-sentence word correlation, while AID shows a high correlation at sentence terminations (indicated by '.'). As highlighted in the yellow box comparison, **FWM, when confronted with unseen subjects (*w/ sys-diff* case), shows a decreased correlation between relevant sentences and an increased correlation among irrelevant ones.** On the other hand, **AID maintains consistent results irrespective of systematic differences.**
>
> Furthermore, we explore similarity patterns between *roles* and *unbinding operators*, as done in [2]. We utilize the *roles* generated at each time step of the input sequence and the *unbinding operators* generated at "?" for each of the read heads ($N_r=3$). Figs. 12 and 13 reveal that both models exhibit a high correlation at the end of each sentence ("."). As seen from the yellow box, the **FWM struggles to link the "." of the question-related sentences in the *sys-diff* case, which may explain the prediction of an incorrect answer** ("office"). In contrast, **the AID shows consistent patterns and accurately predicts the correct answer**.
>
> For the Sort-of-CLEVR task, the baseline model implementation (CNN encoder and the multiple attention layers) makes it hard to analyze the qualitative analysis. So, we only conducted the qualitative analysis of the bAbI task in this review period.
>
> [2] Schlag, I., Munkhdalai, T., & Schmidhuber, J. (2020). Learning associative inference using fast weight memory. *arXiv preprint arXiv:2011.07831*.
>
>
> ***
>
> > For bAbI, a more detailed breakdown of the performance by question type or story length will also be helpful.
>
> We have also included the detailed bAbI results in the Appendix (please see Table 14).

---

> > ### Comment · Reviewer_rEt9 · 2023-12-04
> > **Thank you for addressing the review**
> >
> > Dear authors, thank you for addressing the comments in my review.
> >
> > I appreciate the additional experimental results over the bAbI task, both the analysis per question type on table 14 and the new attention maps -- these look great. Expanding the discussion in the related work section about how the approach compares to slot attention is useful too. Finally, I appreciate the detailed response to my review and to those of other reviewers.
> >
> > While I agree with reviewer bmNb that novelty-wise, the technical difference between slot attention and the paper is not large, nevertheless, for the usefulness of applying it in the context of TPRs and, most importantly, for the thoroughness of the rebuttal and paper updates, including performing additional experiments following the reviewers' comment, I'm happy to raise my score.

---

### Official Review · Reviewer_8A4n · 2023-11-09

**Soundness:** 3 good
**Presentation:** 3 good
**Contribution:** 4 excellent
**Rating:** 8
**Confidence:** 4

**Summary:**

Background information: TPRs are an approach for representing compositional structure in vector space; they work by encoding a compositional structure via pairs of fillers - the components of the structure - and roles - the positions of the fillers in the structure. For instance, in the sentence “cats chase dogs”, the fillers could be the words “cats”, “chase”, and “dogs”, and the roles could be “subject”, “verb”, and “object”, respectively. Each filler and each role is represented with a vector, and these vectors are then combined via tensor products and matrix addition to produce a representation for the whole compositional structure.

What this paper does: The authors introduce an approach called Attention-based Iterative Decomposition (AID) designed to generate role and filler representations for models based on Tensor Product Representations (TPRs). TPRs require the input to be broken down into fillers and roles (both represented as vectors), and this is what AID is designed to do; it can be plugged into any TPR-based system as a way to produce the fillers and roles, which can then be processed in the way they normally are for TPRs. AID starts with an initial proposal for the values of the role and filler vectors. These values are then iteratively updated; at each iteration, each TPR component (i.e., role or filler) attends to the input elements, and the TPR components compete with each other for which component attends to which input element. The result of the attention process at each iteration is a new proposal for the role and filler vectors, which is then the input to the next iteration, until the iteration finishes and the final role and filler vectors are produced. The authors then run experiments where they test 3 TPR-based systems from prior work on 4 tasks that involve systematic generalization. They find that adding the AID module improves compositional generalization across tasks.

**Strengths:**

- S1: The paper addresses an important problem - namely, how to get neural networks to produce effective compositional representations.
- S2: The proposed AID module is intuitive and can act as a drop-in module in any TPR-based system, meaning that it will be straightforward for other authors to adopt.
- S3: AID shows very strong performance in the experiments, often substantially increasing accuracy over previous approaches.
- S4: The experiments are extensive, providing compelling evidence for the strength of the approach.
- S5: In addition to the experiments based on accuracy, there are also analyses of the structure of the learned representations, which deepen the analyses and lend insight into the ways in which the AID module is enhancing the representations.

**Weaknesses:**

- W1: I believe there is a potential confound of number of parameters. That is, if I understand correctly, AID adds more parameters to the model. Therefore, it’s possible that the improvements created by AID are due to having more parameters rather than due to the effectiveness of the strategy. For most of the experiments, the difference in performance is so large that it’s probably not solely due to number of parameters, but for the Wikitext experiment, the improvement that AID brings is pretty small, so it does seem like a more important concern there. The most convincing way to address this concern would be have the same number of parameters in the model version that has AID and the model version that doesn’t; this could be achieved by, for example, making the feedforward size a bit smaller in the AID version than the non-AID version.
- W2: I believe that the paper mischaracterizes the binding problem. The binding problem is the question of how different attributes of a structure can be appropriately bound together; for example, given an image with a red square and a blue triangle, how can a system appropriately associate (that is, bind) colors and shapes in order to represent the fact that you have a red square and a blue triangle, rather than a blue triangle and a red square? There is a separate problem that I’ll call the “decomposition problem” (I don’t think this is a standard term, but it will be useful for this review), which is how to decide what the attributes of a structure are. The paper seems to use the term “binding” or “binding problem” when in fact what it talks about is the decomposition problem. Specific places where this occurs are in the abstract (“because of the incomplete bindings”, “can effectively improve the binding”), the second paragraph of the intro (“these works still have a binding problem”), and the first section of related work (“Binding problem”). The reason why I think that this work is not really about binding is that the part of the TPR formalism that does binding is the step where tensor products are used to combine fillers and roles; the AID module does not alter that portion of the formalism, which is why, properly speaking, I believe it is really about decomposition rather than binding. I would recommend updating the wording to clarify this point.
- W3: I think the paper is not as careful as it should be at distinguishing facts (things that have been empirically demonstrated), goals (things that the authors want to achieve), and plausible guesses (things that we think are likely to be true but can’t be certain of). I would recommend rewording the paper to be more careful about these points; as it stands, some points are presented as facts when I believe they are in fact goals or plausible guesses, and this could potentially mislead readers about how clearly these points have been demonstrated. Here are the specific points that stood out to me:
    - The intro says “these works still have a binding problem … because the decomposition mechanism they employed relies on a simple MLP, which is known to be not effective in capturing the compositional nature of data.” I think that this is plausible but not something that can definitively be stated as a fact; a way to more clearly state what is known vs. not known would be “we find that these approaches still show some difficulties on compositional generalization, likely because the decomposition mechanism they employed relies on a simple MLP, which may not be sufficiently structured to learn the compositional nature of the data.” Specific motivations for these edits: adding “likely” to signal that this explanation is plausible but can’t be definitively said to be the cause; add “may” for a similar reason; changing “capture” to “learn”, because an MLP can capture anything (it’s a universal function approximator), so the actual difficulty would be when it needs to learn something.
    - At the start of section 2, I think the word “effectively” should be removed from “we illustrate how the AID module effectively decomposes”. This section doesn’t show that the AID is effective - that is not demonstrated until later, when there are empirical results. Similarly, near the top of page 4, I would remove the word “effectively” again; I don’t think this work demonstrates that competitive attention on its own is effective at decomposing (as opposed to competitive attention being effective when used in combination with the rest of AID, such as the appropriate initial_components).
    - At the start of section 3, and at the end of “Disentanglement analysis” under 3.1.1, I would recommend removing the word “consequently”. That word asserts a causal connection that has not been demonstrated (we know that the model gets better disentanglement and better task performance, but we can’t be certain that one causes the other); a more valid way to phrase this would be “These results demonstrate the AID module’s efficacy in capturing underlying factors during TPR component generation, which may explain why the AID improves task performance.”
    - Near the top of page 7, it says that AID generates “more accurate representations.” I don’t think that “accurate” is the right word here; a better phrasing might be “representations that better conform to the formal requirements that Smolensky established for ideal TPRs”
- W4: Some aspects of the experimental setup were not clear to me. First is what the input features are; see Q1 below. Second is that I found it somewhat difficult to understand exactly how the tasks worked; this concern could be addressed by providing some examples of the tasks (ideally in the main paper, or in an appendix if there isn’t room). For example, it’s not clear to me what the inputs are in the SAR task - is it just one x and one y? Or a sequence of x’s and y’s, and if so how are they arranged - x y x y x y, or x x x y y y? And how is the model presented with an x?

Overall, I really enjoyed this paper, but I do think that these concerns currently decrease the paper’s understandability as well as the confidence that we can place in its results. If these concerns are addressed, I would be open to raising my score.

UPDATE: The author response has sufficiently addressed my concerns. Therefore, I have raised my score from 5 to 8; previously I said 5 because my concerns prevented me from recommending acceptance, but with these concerns addressed I believe that the paper's helpful contributions are now effectively highlighted, enabling me to happily recommend acceptance.

**Questions:**

- Q1: What are the input features? I was unable to figure this out. Specifically, is each input feature one object (such as one word)? Or is it one element within a vector representation? From most of the paper I was assuming that it was one object. However, I normally associate the word “feature” with an element of a vector representation. Also, if the input features are objects, that makes me confused about why N_inputs is considered a hyperparameter in Figure 5, since that’s really a property of the task rather than a hyperparameter. One thing that could help to clarify this is to show an actual example from an actual task in Figure 1, so that we can see what wach input feature is in the context of that task.
- Q2: Near the top of page 4, it says that producing initial_components from the concatenated input features assigns symbolic meanings to each component, such as roles and fillers. I don’t understand how it achieves this. I can understand why this would be useful (because it would provide a better/optimized starting point), but I don’t see how it pushes each component to have a particular symbolic meaning such as “roles” or “fillers”

---

> ### Author Response · Authors · 2023-11-19
>
> We thank the reviewer for their helpful comments. Also, we updated the manuscript to reflect review comments and marked revision with red color. Our responses to your concerns are as follows:
>
> > W1: I believe there is a potential confound of number of parameters. That is, if I understand correctly, AID adds more parameters to the model … this could be achieved by, for example, making the feedforward size a bit smaller in the AID version than the non-AID version.
>
> Thank you for your feedback. In response to your query and Reviewer bmNb's, **we performed experiments with parameter-increased models on the WikiText-103 task.** In our experiments, we increased the size of the feed-forward network in the attention part of the baseline model. The size increase is applied to the exact same positions of the model where AID is adopted, for a fair comparison with our AID-assisted network architecture. In the Table below, **the experimental results show that improvements obtained with AID do not merely come from the number of increased parameters in the models**.
>
> | **WikiText-103 task** | Valid | Test | # params |
> | --- | --- | --- | --- |
> | Linear Transfomer | 36.473 | 37.533 | 44.02M |
> | Linear Transfomer (more params) | 36.452 | 37.306 | 44.22M |
> | Linear Transfomer (+ AID) | **36.159** | **37.151** | 44.16M |
> | Delta Network | 35.640 | 36.659 | 44.03M |
> | Delta Network (more params) | 35.468 | 36.639 | 44.23M |
> | Delta Network (+ AID) | **35.361** | **36.253** | 44.18M |
>
> Moreover, in additional experiments with other tasks, we applied a different method to increase the model parameters, increasing either the hidden or head size of the baseline models. **These experiments also show that the performance enhancements with AID are not simply due to the increased model parameters, rather the network architecture ( multi-layer vs attention-based slots ) could have contributed more**. We updated these results in Appendix E of the manuscript.
>
> | **bAbI task** | $D_\text{LSTM}$ | *w/o sys diff* | *w/ sys diff* | Gap | # params |
> | --- | --- | --- | --- | --- | --- |
> | FWM | 256 | 0.79 | 2.85 | 2.35 | 0.73 M |
> | FWM (more params) | 512 | 0.75 | 2.16 | 1.41 | 1.89 M |
> | FWM (+ AID) | 256 | **0.45** | **1.21** | **0.76** | 1.23 M |
>
> | **Sort-of-CLEVR task** | $N_\text{heads}$ | $D_\text{heads}$ | *Unary* | *Binary* | *Ternary* | # params |
> | --- | --- | --- | --- | --- | --- | --- |
> | Linear Transformer | 8 | 32 | 82.5 | 78.3 | 60.0 | 0.68 M |
> | Linear Transformer (+ AID) | 8 | 32 | **98.9** | 78.0 | 61.0 | 0.83 M |
> | --- | --- | --- | --- | --- | --- | --- |
> | Linear Transformer | 4 | 64 | 69.3 | 75.5 | 56.4 | 0.68 M |
> | Linear Transformer (+ AID) | 4 | 64 | **98.9** | **78.6** | **63.7** | 0.83 M |
> | --- | --- | --- | --- | --- | --- | --- |
> | Linear Transformer | 8 | 64 | 57.5 | 59.7 | 53.2 | 2.55 M |
> | --- | --- | --- | --- | --- | --- | --- |
> | Linear Transformer | 4 | 128 | 57.9 | 59.9 | 52.2 | 2.55 M |
>
>
>
> ***
>
>
> > W2: I believe that the paper mischaracterizes the binding problem … I would recommend updating the wording to clarify this point.
>
> Thank you for your valuable input. Initially, we used the term "binding problem" based on its multifaceted description in [1], which outlines three key perspectives: (a) segregation, involving the decomposition of low-level sensory data into distinct entities; (b) representation, referring to the separation of information at a representational level; and (c) composition, which pertains to the formation of new inferences. TPR explicitly provides a representational separation to neural networks in the form of *role* and *filler*. Based on this property, TPR-based models have shown significant improvements in systematic generalization, but still have the potential risk if they fail to find correct structured representations from data. Therefore, our focus was primarily on the "segregation" aspect of the binding problem. However, as you pointed out, we recognize that the term "binding problem" could mislead readers. To clarify our focus and address your concerns, **we change our expression of "binding problem" to "decomposition problem".** As you suggested, this change aims to avoid misunderstandings of the reader for our target problem.  Furthermore, the term "decomposition" is also closely aligned with the concept of "r/f decomposition" mentioned in the foundational TPR paper [2]. Following this change in terminology, we accordingly updated **every sentence in our manuscript, whenever “**binding problem**” is used to** mean the "segregation" aspect of the problem [Abstract, Second paragraph of Section 1, First paragraph of Section 2 (Originally, Section 4)]
>
> [1] Greff, K., Van Steenkiste, S., & Schmidhuber, J. (2020). On the binding problem in artificial neural networks. *arXiv preprint arXiv:2012.05208*.
>
> [2] Smolensky, P. (1990). Tensor product variable binding and the representation of symbolic structures in connectionist systems. Artificial intelligence, 46(1-2), 159-216.

---

> ### Author Response · Authors · 2023-11-19
>
> ***
>
> > W3: I think the paper is not as careful as it should be at distinguishing facts (things that have been empirically demonstrated), goals (things that the authors want to achieve), and plausible guesses (things that we think are likely to be true but can’t be certain of) … Here are the specific points that stood out to me
>
> In response to your feedback, we have rephrased the highlighted sentences. The revised sentence is as follows:
>
> We remove the word "effectively". [First paragraph of Section 3 (originally, Section 2), Fourth paragraph of Section 3.2 (originally, Section 2.2)]
>
> **Sentence 1)** However, these works still have a binding problem … the compositional nature of data.
>
> $\rightarrow$ **Rewording 1)** However, we find that these approaches still encounter challenges in achieving compositional generalization. This is likely attributable to their reliance on a simple MLP for decomposition, which may not be structured to learn the compositional nature of data. [Second paragraph of Section 1]
>
> **Sentence 2)** These results demonstrate …, consequently improving task performance.
>
> $\rightarrow$ **Rewording 2)** These results demonstrate the AID module’s efficacy in capturing underlying factors during TPR component generation, which may explain why the AID improves task performance. [Third paragraph of Section 4.1.1 (originally, Section 3.1.1)]
>
> **Sentence 3)** From all these results, it is clear that the AID module learns to decompose data into meaningful components and generate more accurate representations than the baseline model.
>
> $\rightarrow$ **Rewording 3)** From all these results, it is clear that the AID module learns to decompose data into meaningful representations that better conform to TPR conditions than the baseline model. [Fifth paragraph of Section 4.1.1 (originally, Section 3.1.1)]
>
>
> ***
>
> > W4: Some aspects of the experimental setup were not clear to me … And how is the model presented with an x?
>
> Thank you for pointing out the need for a clearer task description in our manuscript. Recognizing this, we have taken steps to enhance the reader's understanding. Specifically, **we have added an illustrative figure (Fig. 6) to visually depict the SAR task process.** Additionally, **a detailed explanation of how the SAR task operates has been included in Appendix A.1.** For further clarification, as follows. At each training iteration, generative factors ($x$ and $y$) are sampled from word sets $X$ and $Y$ to construct the input sequence. The sampled $x$ and $y$ values are randomly paired, creating combinations of one $x$ and one $y$ each. These pairs are then embedded into a vector space and concatenated with flags, which are scalar values signaling the start of the discovery and inference phases. In the discovery phase, models sequentially receive these concatenated representations. During the inference phase, the model is presented only with $x$ values (considered as *role* information) and is tasked with predicting the corresponding $y$ values (considered as *fillers*).

---

> ### Author Response · Authors · 2023-11-19
>
> > Q1: What are the input features? … One thing that could help to clarify this is to show an actual example from an actual task in Figure 1, so that we can see what each input feature is in the context of that task.
>
> Thank you for your feedback. We have updated Figure 1 to provide a clearer understanding. We set up the input feature differently based on which type of baseline model the AID is integrated with. TPR-RNN generates TPR components for each sentence, and thus we designate each word as an individual input feature in TPR-RNN. On the other hand, FWM and Linear Transformer generate the components for each word (or token). To maintain the operations of baseline models, we simply partition the word (or token) vector into $N_\text{inputs}$ sub-vectors of dimension $D_\text{inputs}$ and designate each sub-vector as an individual input feature in FWM and Linear Transformer. Therefore, the input features are word vectors in the sentence-level model while partitioned sub-vectors in the word(or token)-level model.
>
> ***
>
> > Q2:  Near the top of page 4, …  but I don’t see how it pushes each component to have a particular symbolic meaning such as “roles” or “fillers”
>
> Thank you for highlighting the need for a more comprehensive explanation. We acknowledge that the description provided in **the fourth paragraph of Section 4.2 (Originally, Section 3.2)** did not sufficiently explain how the initial_components of our model are assigned symbolic meanings. To address this, **we have revised the text to enhance clarity and understanding.**
>
> We point out that the iterative attention mechanism, in its pure form, generates permutation-invariant components, posing a challenge in directly assigning the representations as elements in TPR functions (e.g., should identify the appropriate slot for *filler*). To navigate this challenge, we propose a trainable routing strategy for integrating attention and TPR. This method systematically associates each initial component with a specific symbolic meaning. As illustrated in Fig. 1, for instance, the first component is designated as *role1* and the third as *filler*. Specifically, these initial_components are obtained by applying a feed-forward network to the concatenated input features and are optimized to facilitate a more effective decomposition during training. **Through this routing strategy, the AID learns to generate the structural representations for TPR.**

---

> > ### Comment · Reviewer_8A4n · 2023-11-22
> > **Thank you for the response; raised score**
> >
> > Thank you for your detailed reply! These responses sufficiently address the main concerns that I had: The results with increased parameter counts are convincing; the term "decomposition problem" is clear; and the other wording changes are also clear. Therefore, I have raised my score from 5 ("marginally below acceptance threshold") to 8 ("accept good paper").

---

> > > ### Author Response · Authors · 2023-11-23
> > > **Thank you for your time**
> > >
> > > We thank Reviewer 8A4n for valuable comments and thank for raising the score. We are glad our updates successfully addressed your comments and clarified some ambiguities.

---

### Author Response · Authors · 2023-11-19
**Summary of Revision**

We appreciate all reviewer's constructive comments and feedback. We updated our paper according to the reviewer's concerns as follows.

- Update Figure 1 to depict input features `[Response to Reviewer 8A4n]`

- Relocate the Related work section to precede the Method section `[Response to Reviewer rEt9]`

- Rewrite the description of the initial components in Section 3.2 `[Response to Reviewer 8A4n]`

- Rewrite the WikiText-103 section to better illustrate our intention `[Response to Reviewer rEt9 and bYLE]`

- Reword the "Binding problem" to the "Decomposition problem" in the Abstract, Introduction, and Related work `[Response to Reviewer 8A4n]`

- Add more discussion with prior work in Related work `[Response to Reviewer rEt9]`

- Add an illustrative figure (Fig. 6) to depict the SAR task in Appendix A.1 `[Response to Reviewer 8A4n]`

- Add additional experimental results (Fig. 9 and Tables 11, 12, and 13) with baseline models with more parameters in Appendix E `[Response to Reviewer 8A4n and bmNb]`

- Add additional qualitative analysis (Figs. 10, 11, 12, and 13) on the bAbI task in Appendix F `[Response to Reviewer rEt9]`

- Add the detail of bAbI results (Table 14) in the Appendix `[Response to Reviewer rEt9]`

---

### Meta-Review · Area_Chair_uo9N · 2023-12-04

**Metareview:**

This paper contributes the Attention-based Iterative Decomposition (AID) module, which generates role and filler representations for models based on Tensor Product Representations (TPRs). It effectively addresses the binding problem of segregation (or perceptual grouping) to facilitate structured tensor-product representations. The proposed AID module is quite similar to Slot Attention (Locatello et al., 2020), which fulfills a similar purpose in decomposing inputs to obtain symbol-like representations (eg. of object in images). Nonetheless, there are differences such as the distinction between role and fillers that information is being routed to; and getting rid of permutation equivariance. AID is shown to enhance the capabilities of existing TPR-based models, which have shown to be particularly suitable for systematic generalization in (natural) language domains. AID is analyzed in detail, which contributes to its understanding.

Initially this work received mixed reviews slightly in favor of rejection, despite its strengths. In particular, although reviewers noted that this work addresses an important problem, that the AID module is intuitive and well-designed, and commended the extensive experiments, there were also concerns. For example, multiple reviewers point out that AID is highly similar to Slot Attention, yet this was sufficiently clear from the paper. Further, there were concerns regarding the fairness of the comparison to baselines in terms of capacity, and qualitative evaluation. The authors have done a good job addressing many of the outstanding concerns and in response multiple reviewers have increased their score. I note that two reviewers provided reviews with a confidence of 2, which is very low and have therefore not been taking into consideration fully.

At this point the remaining outstanding concerns are significance and novelty, i.e. the paper’s technical novelty is limited. and even though it contributes useful insights and detailed experiments for deploying TPR-based methods only a smaller community will likely benefit from them. On the other hand, within said community there is clearly excitement and as one of the reviewers comments: “I truly believe that this mechanism could help me in my research, and I am eager to send this paper to my collaborators once it is de-anonymized because I believe it could be useful to them too”. I also agree that the latter is less of a concern and that the paper is insightful even for others outside this subfield. In the end, my recommendation is to accept this paper.

For the camera-ready, aside from integrating all the reviewer comments, I would further encourage the authors to point out similarities to Slot Attention even before the related work section (eg. Introduction, etc.), since it is such a core part of AID and something that multiple reviewers struggled with. In fact starting the discussion of AID from Slot Attention might be even more appropriate.

**Justification For Why Not Higher Score:**

Given the technical novelty and smaller target audience I don't think the community would benefit from having this be a spotlight paper.

**Justification For Why Not Lower Score:**

This paper is a good example of taking something that works (here: Slot Attention) and adapting it (here: proposing some key changes to the algorithm to work with roles/fillters) to a different domain (here: learning role-filler representations for TPR-based language processing) to get improvements (here: in terms of improved systematic generalization).

The experiments and analyses are thorough and I am quite confident that conference attendees could benefit from exposure to these ideas and results, even if they are not directly working with TPR-based methods.

---

### Decision · Program_Chairs · 2024-01-16

Accept (poster)